# A real-time regional model for COVID-19: Probabilistic situational awareness and forecasting

**Solveig Engebretsen**[1], **Alfonso Diz-Lois Palomares**[2], **Gunnar Rø**[2], **Anja Bråthen Kristoffersen**[2], **Jonas Christoffer Lindstrøm**[2], **Kenth Engø-Monsen**[3], **Meghana Kamineni**[4], **Louis Yat Hin Chan**[2], **Ørjan Dale**[5], **Jørgen Eriksson Midtbø**[2,5], **Kristian Lindalen Stenerud**[5], **Francesco Di Ruscio**[2], **Richard White**[2], **Arnoldo Frigessi**[4☯], **Birgitte Freiesleben de Blasio**[2,4☯]\***

**1** SAMBA, Norwegian Computing Center, Oslo, Norway, **2** Department of Method Development and Analytics. Norwegian Institute of Public Health, Oslo, Norway, **3** Telenor Research, Fornebu, Norway, **4** Oslo Centre for Biostatistics and Epidemiology. University of Oslo and Oslo University Hospital, Oslo, Norway, **5** Telenor Norge AS Fornebu, Norway

☯ These authors contributed equally to this work.
\* Birgitte.Freiesleben.DeBlasio@fhi.no

**Data Availability Statement:** Individual requests for the regional COVID-19 case data should be made to Helsenorge.no; information is available at https://www.fhi.no/en/more-access-to-data/

## Abstract

The COVID-19 pandemic is challenging nations with devastating health and economic consequences. The spread of the disease has revealed major geographical heterogeneity because of regionally varying individual behaviour and mobility patterns, unequal meteorological conditions, diverse viral variants, and locally implemented non-pharmaceutical interventions and vaccination roll-out. To support national and regional authorities in surveilling and controlling the pandemic in real-time as it unfolds, we here develop a new *regional* mathematical and statistical model. The model, which has been in use in Norway during the first two years of the pandemic, is informed by real-time mobility estimates from mobile phone data and laboratory-confirmed case and hospitalisation incidence. To estimate regional and time-varying transmissibility, case detection probabilities, and missed imported cases, we developed a novel sequential Approximate Bayesian Computation method allowing inference in useful time, despite the high parametric dimension. We test our approach on Norway and find that three-week-ahead predictions are precise and well-calibrated, enabling policy-relevant situational awareness at a local scale. By comparing the reproduction numbers before and after lockdowns, we identify spatially heterogeneous patterns in their effect on the transmissibility, with a stronger effect in the most populated regions compared to the national reduction estimated to be 85% (95% CI 78%-89%). Our approach is the first regional changepoint stochastic metapopulation model capable of real time spatially refined surveillance and forecasting during emergencies.

applying-for-access-to-data/. Nonsensitive Norwegian COVID-19 case data until November 2022 are available at: https://github.com/folkehelseinstituttet/surveillance_data/tree/master/covid19/. Additional example code for calibration, estimating hospitalisation parameters and model validation is available at https://github.com/folkehelseinstituttet/realtime-regional-model. Requests for access to mobility data should be directed to Telenor Research by email: TelenorResearch@telenor.com. A simulated mobility data set can be found at https://github.com/geirstorvik/smc.covid. The infectious disease spread model is publicly available on GitHub in the R-package spread and can be found at https://github.com/folkehelseinstituttet/spread.

**Funding:** This work was supported by the Research Council of Norway Grant No. 312721 (SE, ADLP, GR, ABK, JCL, KEM, LYHC, JEM, FDR, RW, AF, BFdB) and Grant No. 237718 (SE, AF) and the Nordic Research Council Grant No.105572 (AF). The funders had no role in study design, data collection and analysis, decision to publish, or preparation of the manuscript.

**Competing interests:** The authors have declared that no competing interests exist.

## Author summary

National and regional governments have imposed restrictions on their citizens to control the spread of the SARS-CoV-2 virus, which caused disruption of their lives. This pandemic has been geographically and temporally extremely complex and heterogeneous, difficult to understand and predict. Important decisions assumed homogeneity across local communities, regions and time, because of a lack of instruments to rapidly capture regional and temporal differences. We present a complete probabilistic model to estimate regional processes in space and time, focussing on operational usefulness in real time. It uses real-time mobile phone mobility data, laboratory-confirmed cases, data on cases imported from abroad, and hospitalisation incidence. We propose a new calibration method to estimate regional reproduction numbers, which handles the high dimensional parameter space. The model has been successfully used for local situational awareness and forecasting in Norway, contributing to one of the most successful handlings of the epidemic in Europe. We estimate both national and region-specific reduction effects of the lockdown. We find larger reduction effects in the more populous regions compared to the national average, where we estimate a reduction of 85% in the transmissibility (95% CI 78%-89%).

## Introduction

The COVID-19 pandemic has led to an unprecedented global crisis. Health authorities worldwide are continuously striving to effectively mitigate the disease spread, balancing the protection of health with social and economic costs. In most countries, the disease spread is characterised by significant geographical and temporal heterogeneity, requiring local tailoring of interventions. Therefore, time-sensitive information about the regional conditions becomes essential to management.

Mathematical modelling has been influential for preparedness planning and decision-making [1–6]. Most of the published COVID-19 models are at national scale and when at regional level they do not account for inter-regional mobility [7–11].

We developed a real-time spatio-temporal SARS-CoV-2 metapopulation model for assessment, monitoring, and short-term prediction, to inform regional and national policy decisions. The model exploits mobile phone mobility data and is calibrated to region-specific daily hospital incidence and laboratory-confirmed cases using a new sequential Monte Carlo Approximate Bayesian Computation (SMC-ABC) [12] which we call the Split-SMC-ABC.

Norway is among the countries with the lowest COVID-19 death rate in Europe [13]. The disease spread in Norway is characterised by continuous high circulation in Oslo and the densely populated areas surrounding the capital, and spatially shifting outbreaks in the remainder and less occupied regions of the country. Before vaccination, the Norwegian mitigation strategy focussed on early detection, isolation, and municipality-based contact tracing, supported by national contact tracing teams, for rapid quarantining of contacts. Norway experienced its first confirmed SARS-CoV-2 cases in February 2020, followed by a rapid acceleration of cases. On March 12, 2020, the Norwegian government enforced a nationwide lockdown [14], succeeded by border closure [15] and internal travel restrictions [16]. In the following months, the measures were gradually removed. The epidemic resurged in the autumn and winter, leading to new national restrictions [17–19] and many locally targeted interventions [20–22].

By including geography in our transmission model, we can provide surveillance indicators and predictions on a regional level which can take into account different interventions in different regions. Regional estimates are essential for local decision making, but they arise from a much more difficult high-dimensional estimation task. Our proposed calibration method has been designed to handle the increasing number of parameters over time, so to provide timely estimates of key model-derived indicators, including reproduction numbers for situational awareness, and prediction of future hospital and intensive care unit admissions.

Although our model is tailored to the management of the early COVID-19 epidemic, it remains applicable as a situational awareness tool during vaccination roll-outs. The methodology can be applied to other countries and settings, and tailored to other infectious diseases where regional indicators need to be inferred as the epidemic unfolds.

In this study, our objective was three-fold: (i) to present the new regional model and our inferential method, (ii) to report on the weekly epidemiological situation during 2020–2021 in Norway, where the model was used to inform real-time policy decisions, and (iii) to compare and validate our modelling approach. The article is organised in the following way. First, we provide a technical presentation of our regional model, including a model for using real-time mobility data, and a novel split-ABC-SMC inference technique. In the result sections, we document the implementation of the model with Norwegian data and report on county-specific effective reproduction numbers, alongside national results obtained from an alternative model ("null model"), excluding regional differences in the transmissibility. We present short-term forecasting results of hospital admissions and test data for the counties, obtained at different pandemic stages, covering highly volatile and relatively stable periods. Finally, we document the enhanced predictive performance of the regional model by comparing its projections to those obtained from a national null model and a simple regional model forecasting from the past two-weeks data.

## Materials and methods

### Ethics statement

The ethical approval for the use of data in this article was given under the Norwegian Health Preparedness Act, paragraphs 2–4, more information is available at https://www.fhi.no/en/id/infectious-diseases/coronavirus/emergency-preparedness-register-for-COVID-19/.

### Case data

This study utilises different COVID-19 data sources. Beredskapsregisteret for COVID-19 (Beredt C19) gathered all the Norwegian COVID-19 data and made it available for use. In Norway, the use of a unique personal identifier makes it possible to link different data sources. We obtained anonymised individual-level hospital incidence data from the Norwegian Intensive Care and Pandemic Registry (NIPaR). This data set included all patients admitted within 14 days of a positive laboratory test or diagnosed with COVID-19 in the discharge report, by April 12, 2021. The data set contained age, residence region, date of hospitalisation, date of entering mechanical respirator, and discharge dates from both mechanical respirator and hospital. NIPaR was merged with data from the Norwegian Surveillance System for Communicable Diseases (MSIS), which comprises data on all notifiable infectious diseases in Norway, including COVID-19. The MSIS data included age, whether the patient was infected abroad or in Norway, date of positive test, and date of symptom onset for all COVID-19 cases. We used the data both to calibrate the transmissibility in the model and to estimate the hospitalisation parameters.

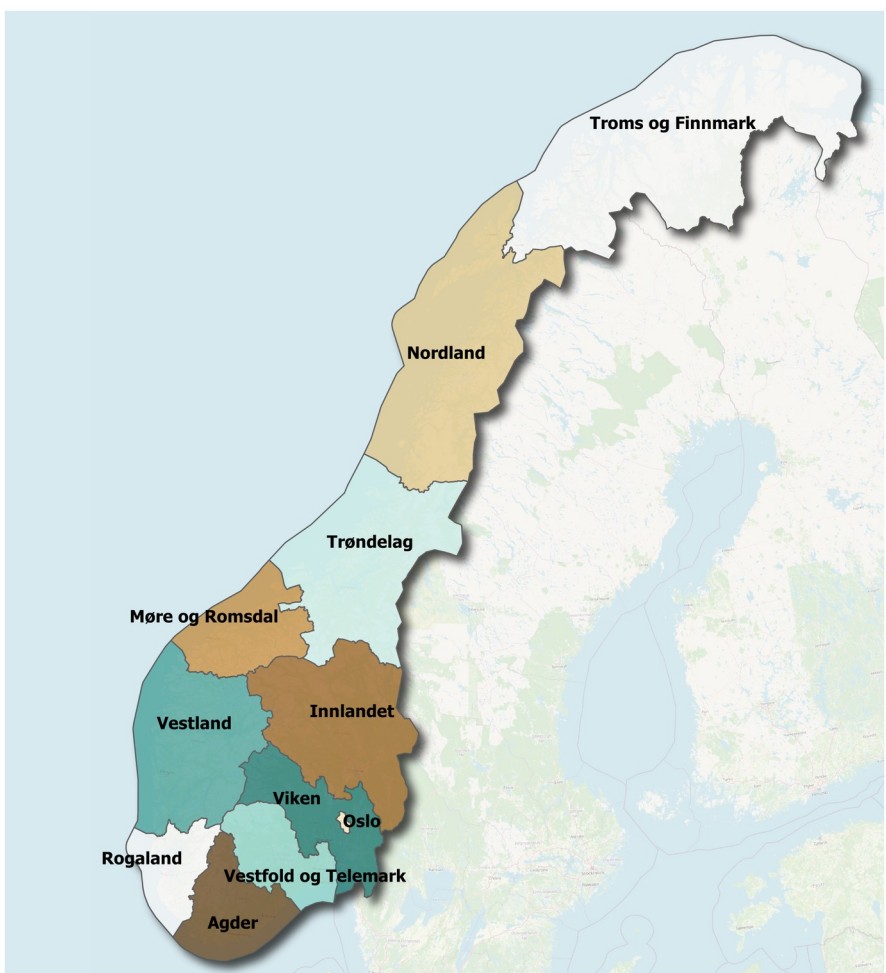

**Fig 1. Norwegian geography.** Overview of Norway's 11 counties. The map was made using an open-source shape file with License Creative Commons BY 40 (CC BY 4.0) from Kartverket (https://kartkatalog.geonorge.no/metadata/administrative-enheter-fylker/6093c8a8-fa80-11e6-bc64-92361f002671), and background from OpenStreetMap overview (https://www.openstreetmap.org).

We obtained data from MSIS on the number of negative SARS-CoV-2 tests by region from April 1, 2020, when the reporting commenced. In this data set, negative tests within seven days of a previous negative test were excluded.

In the calibration, we use county level resolution of the surveillance data. An overview of the 11 counties in Norway is provided in Fig 1. The counties are further divided into 356 municipalities, but in the case of Norway we have chosen to simulate on the county level. In the paper, we will use the term region when we refer to the general model and methodology, and county when we refer to the case of Norway.

## Mobility data

The geographical spread of COVID-19 is governed by movements of infectious individuals. We used mobile phone mobility data from the operator Telenor Norway as a measure of mobility of individuals between municipalities in real-time. Every mobile phone is continuously connected to a cell tower and switches from one tower to another as it moves between

their areas of connectivity. The geographical location of the connected cell tower provides an approximate location of the phone, and therefore of its owner. For every six-hour time period, each individual subscriber of the mobile operator was assigned to the municipality most recently visited in the last two hours of the six-hour time window. An individual had moved from municipality $i$ to municipality $j$ if they were in $i$ during the last two hours of one six-hour period and in $j$ in the final two hours of the next six-hour period. The population travel patterns were then described by counting how many subscribers transitioned between any pairs of municipalities, during two consecutive six-hour periods. This represents an origin-destination mobility matrix. Each mobility matrix contains aggregated counts of people travelling, anonymised by aggregation in space and time. There are no individual identifiers in the data. The counts in the generated mobility matrices were further up-scaled by a factor representing the overall Telenor Norway market share, estimated at 47.5% in 2019 [23]. As a final privacy-preserving precaution, counts below 20 in the mobility matrices were censored to prevent potential re-identification of individuals or small groups of people. The cell tower network also connects to every device and handset in the mobile operator's network. To focus only on people movements, we have filtered out devices (IoT/M2M) that are not likely to be carried by an individual.

The time series of mobility matrices covered the period from the first confirmed case in Norway up to April 1, 2021. In this way, the model was informed by real-time mobility patterns since the first confirmed case in Norway. We utilised actual mobility data when simulating the past. In this way, alterations in the mobility patterns as a response to interventions, seasonality, or other causes, were automatically incorporated in our model predictions. While the mobile phone mobility data was on municipality level, we aggregated the data to county level in the simulations.

The model used a regularised version of the mobility matrix of the most recent non-holiday weekday when predicting future events. First, the four mobility matrices for the given weekday were averaged, i.e. the four six-hourly matrices for each 24 hour-interval. Then an optimisation was performed to obtain the closest matrix that preserved population. The technical details of the regularisation is provided in S1 Text, section S1.1. The regularisation dates used in the predictions are provided in S1 Text, Table A.

## Regional transmission model

A regional metapopulation model was defined to describe the spread of COVID-19 in space and time. The model was an extension of the SEIR metapopulation model [24] and consisted of three layers: (i) the population structure in each region, based on census data (in the case of Norway from Statistics Norway [25]), (ii) a dynamic network of movements describing the travel patterns between regions (using the Telenor mobility matrices), and (iii) a local transmission model within each region.

The population was initialised according to data on population sizes for the regions (for Norway in January 2020 [25]). Each individual in the model then had a designated home location. To model the disease spread, the individuals first mixed in their current location for six hours. Then, the individuals travelled according to the mobility matrices, and then mixed again in their new location for six hours. When we implemented the travel from location $i$ to $j$, we preferentially sent back the individuals with home location $j$ who were visitors in location $i$. If more people travelled from $i$ to $j$ according to the mobility matrices than those from $j$ currently in $i$, we sent individuals who belonged to $i$ and were currently present in location $i$. If this was still not enough, we sent to $j$ other visitors in $i$, who belonged to other regions. In this way, the virus could spread from one region to another, for example by an individual travelling

from a susceptible location to an infected location, acquiring the infection, and then travelling back. We preferentially sent individuals back to their home location, keeping track of where they came from, in order not to overestimate the spread [26]. We moved individuals independently of their disease status (i.e. using a multinomial distribution with probabilities based on proportions of individuals in each state). Another option would have been to decrease the probability of travel for e.g. the infectious symptomatic, however we have chosen not to do this, as it would still require ad hoc assumptions on the size of the decreased travel probability. In Norway, local mobility and commuting were of primary importance. The number of infected individuals in a municipality thus changes both with the disease dynamics between each six-hour period, and due to mobility between locations every sixth hour. The local rules which we here adopted can be changed in other situations and countries. In S1 Text, Section S2.7 we provide a sensitivity analysis to the mobility rules used and show that the outcomes are in general robust to the different mobility rules studied.

**Local stochastic epidemiological model.** For each time interval, we assumed homogeneous mixing among the individuals present within each region. We assumed that the individuals could be in one out of six states: susceptible ($S$), exposed and not infectious ($E_1$), presymptomatic and infectious ($E_2$), infectious symptomatic ($I$), infectious asymptomatic ($I_a$), or recovered ($R$). The additional compartments were included to better fit the epidemiology of COVID-19 transmission. Multiple studies indicate the importance of presymptomatic transmission [5, 27, 28] and asymptomatic infection [29]. We implemented the transitions between the states for individuals currently in location $i$ by stochastic difference equations:

$$S^i(t + \delta t) = S^i(t) - Y_1(t), \tag{1}$$

$$E_1^i(t + \delta t) = E_1^i(t) + Y_1(t) - Y_2(t), \tag{2}$$

$$E_2^i(t + \delta t) = E_2^i(t) + Y_3(t) - Y_4(t), \tag{3}$$

$$I^i(t + \delta t) = I^i(t) + Y_4(t) - Y_5(t), \tag{4}$$

$$I_a^i(t + \delta t) = I_a^i(t) + Y_2(t) - Y_3(t) - Y_6(t). \tag{5}$$

The stochastic transitions between the states were

$$Y_1(t) \sim \text{Binom}(S^i(t), \beta_t^i \delta t / N_t^i \cdot (I^i(t) + r_{I_a} I_a^i(t) + r_{E_2} E_2^i(t))),$$

$$Y_2(t) \sim \text{Binom}(E_1^i(t), \lambda_1 \delta t),$$

$$Y_3(t) \sim \text{Binom}(Y_2(t), (1 - p_a)),$$

$$Y_4(t) \sim \text{Binom}(E_2^i(t), \lambda_2 \delta t),$$

$$Y_5(t) \sim \text{Binom}(I^i(t), \gamma \delta t),$$

$$Y_6(t) \sim \text{Binom}(I_a^i(t), \gamma \delta t).$$

Here, $\beta_t^i$ is the probability of transmission upon a contact times the contact rate, $r_{I_a}$ is the relative infectiousness of the asymptomatic, $r_{E_2}$ is the relative infectiousness of the presymptomatic, $1/\lambda_1$ is the latent period, $p_a$ is the probability of being an asymptomatic carrier, $1/\lambda_2$ is the

presymptomatic period, $1/\gamma$ is the infectious and asymptomatic infectious period, assumed equal, and $\delta t$ is the time step of the model, set to six hours in our setting. We had one set of equations for each region $i$. The per-compartment counts $S^i$, $E_1^i$, $E_2^i$, $I^i$, $I_a^i$, and $R^i$ are the number of individuals in each compartment who are currently present in the region. For the transmission parameter $\beta_t^i$, the subscript denotes time-dependence, as we assumed it to be a step-function, changing at different pre-specified changepoints. The superscript indicates region, as we allowed different transmissibility for each region. $N_t^i$ is the number of individuals present at time $t$ in the region $i$, hence

$$N_t^i = N_{t-1}^i + \sum_{j \neq i} X_{ij}^t - \sum_{j \neq i} X_{ji}^t,$$

where $X_{ij}^t$ denotes the number of people moving from location $i$ to location $j$ between time $t$ and time $t + 1$.

We do not provide the equation for $R^i(t)$, the number recovered, as we assumed constant population sizes.

By calculating the largest eigenvalue for the next generation matrix of the corresponding deterministic system of differential equations [30], we find an equation for the basic reproductive number $R_0$ of our epidemiological model, given by

$$R_0 = \beta_0 \cdot ((1 - p_a)/\gamma + p_a r_{I_a}/\gamma + (1 - p_a)r_{E_2}/\lambda_2). \tag{6}$$

We calculated the effective reproduction numbers by multiplying the estimated reproduction number (given by the estimated $\beta_t^i$ instead of $\beta_0$ in Eq 6) by the estimated mean proportion susceptible in the corresponding period. We chose to put our prior distributions on the reproduction numbers instead of the $\beta_t^i$, as we had a better understanding of their size. Then we reversed Eq 6 to calculate the $\beta_t^i$ from the reproduction numbers.

We calibrated two different models for the transmissibility. In one model, we assumed the same transmissibility in all counties, hence estimated national reproduction numbers. In the second model, we allowed county-specific transmissibility parameters and hence region-specific reproduction numbers.

Note that the disease spread model has a time resolution of six hours, while all the surveillance data have a daily resolution. We therefore only used every fourth time step of the model when comparing to data, corresponding to 24-hours between each simulated value.

**Changepoint specification.** For Norway, we estimated the first changepoint nationwide between March 12, 2020 and March 16, 2020 which was when the first national lockdown was imposed. The best fit was for March 15, 2020, see S1 Text, Section S2.4. The second changepoint was set on April 20, 2020, when the kindergartens reopened nationwide. After that, the regional changepoints were set when there were changes in regional policies. In addition, we included further changepoints to capture other potential gradual changes in behaviour or viral properties like the gradual takeover by the more transmissible alpha variant in winter 2021 and potential compliance fatigue with restrictions over time. For example, new viral variants are typically more transmissible, requiring changepoints in our model for the reproduction number to capture the change in transmissibility. See for example [31] for evidence of increased transmissibility of the alpha variant in Norway compared to the previously circulating Norwegian variants.

**Importation of the virus.** We seeded the epidemic with the known infected cases imported from abroad, and located them in their county of residence. As we expect that some imported cases went undetected, we imported an additional random, Poisson distributed number of cases for each observed imported case, with mean estimated from the data during

calibration. For Norway in this paper we calibrated this amplification factor between February 17, 2020 and April 1, 2021.

## Model for admission to hospital and intensive care

Based on the estimated incidence in each region obtained from the spatial metapopulation model, we modelled the number of individuals admitted to the hospital. In the model assuming a common, national transmissibility, we used a binomial distribution with hospitalisation probabilities provided in S1 Text Table B. In the model with region-specific transmissibilities, we instead used a 28-days moving average of the daily estimated hospitalisation risk. We use a moving average because the daily hospital admission data were prone to randomness and noise. This means that in the model with a national transmissibility, we assumed hospitalisation risks constant every calendar month, while when we modelled transmissibility regionally we assumed smoothed daily hospitalisation risks. It would be straightforward to also include daily moving averages in the national hospitalisation model. When simulating, once an individual was selected to be sent to hospital, we generated the delay from onset of symptoms to hospitalisation and the length of stay in hospital from two negative binomial distributions. For Norway, we estimated these parameters from individual-level registry data.

The hospitalisation risks were based on the age-specific estimated hospitalisation probabilities in [32]. Even though the compartmental model did not have age compartments, we needed to consider age when computing hospitalisations, as the probability of requiring hospitalisation was highly dependent on age. We corrected the hospital admission probabilities to account for age-dependencies in transmission, by adjusting for the age distribution of the positive cases. For the period where we do not use the test data (prior to May 1, 2020), we computed the probability of hospitalisation by taking into account the demographic age profile in each region. The details on the delay distributions and hospitalisation risks are provided in S1 Text, Section S1.2.

## Laboratory-confirmed cases

To calibrate to the observed number of laboratory-confirmed cases, we simulated the number of infected individuals detected by testing (positive test cases). We assumed that the daily number of positively tested cases could be modelled as a binomial process of the simulated daily total incidence of symptomatic and asymptomatic cases, with detection probability $\pi(t)$. We also assumed a delay $d$ between entering the infectious symptomatic or asymptomatic class and a positive test. In order to make the hospitalisation data and the test data consistent, we needed to capture the changes in test criteria, and we chose to model this in the detection probability $\pi(t)$, as

$$\pi(t) = \exp(\pi_0 + \pi_1 \cdot k_t)/(1 + \exp(\pi_0 + \pi_1 \cdot k_t)),$$

where $k_t$ in day $t$ is a 7-days backwards moving average of the total number of tests performed (with both positive and negative results), and $\pi_0$ and $\pi_1$ are two parameters that we estimated, assuming $\pi_1 > 0$.

In Norway, the testing criteria and capacity have changed significantly since early in the epidemic. Therefore, we only calibrated to the test data from May 1, 2020. The resulting estimated detection probability is shown in Fig 2. We see that it was increasing until autumn 2020, before becoming more stable.

Note that this is a very simple model. For example, one may argue that the probability of detecting a positive case should also depend on the total number infected, not only through the total number of tests. However, without additional and more specific data able to inform

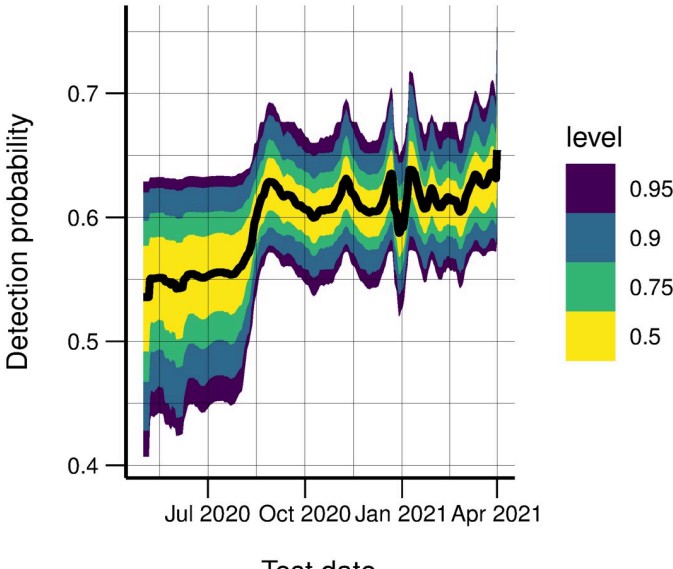

**Fig 2. Detection probability.** Estimated probability to detect a case by testing.

about the detection probability (e.g. seroprevalence data), we were not able to estimate additional parameters.

## Assumptions on model parameters

The parameters related to the natural history of COVID-19 were assigned fixed values or distributions based on the literature. We based the natural history parameters on estimates from [5], with an adjustment of the presymptomatic period based on [28]. Specifically, we assumed a latent period that was exponentially distributed with expected duration of three days, a presymptomatic period that was exponentially distributed with expected duration of two days, an infectious period (time in $I$ and $I_a$) that was exponentially distributed with expected duration of five days, a probability of asymptomatic infection of 40%, a relative infectiousness of presymptomatic of 1.25 compared to the infectiousness in $I$ which is 1, and a relative infectiousness of 0.1 of the asymptomatic cases. Our assumptions are also in agreement with [33], who found an incubation period for COVID-19 of 5.1 days. A complete overview of the parameter choices is provided in S1 Text, Table C.

## Calibration

In the model for Norway with a single national transmissibility, we estimated the transmissibility parameters $\beta_t$ between given changepoints, the parameters $\pi_0$, $\pi_1$, in the expression for the detection probability, the delay $d$ between entering the infectious symptomatic or asymptomatic class and a positive test, and the amplification factor for the imported cases. As of April 1, 2021, this resulted in 19 nationally estimated parameters because of 14 changepoints in the transmission. In the model with regionally varying transmission parameters $\beta_t^i$, we calibrated these region-specific transmissibility parameters instead of national ones. For Norway, as of April 1, 2021, we used in total 94 regional transmissibility parameters, corresponding to fewer changepoints per region than for the national model, as this is a much more difficult estimation problem. For the same reason, we first estimated the amplification factor, $\pi_0$, $\pi_1$, and $d$ in

the national calibration and then used the obtained posterior distributions of these parameters as priors for the same parameters in the regional calibration. The assumed prior distributions for the other parameters are provided below.

We developed a new sequential Monte Carlo version of Approximate Bayesian Computation (SMC-ABC) to estimate the parameters of our model. The idea of ABC is to identify parameters that produce simulated data that are close to the observed data, measured in some distance. We used the least squares between the simulated regional daily hospital incidence and the observed regional daily hospital incidence, summed over all the regions. Similarly, we calculated the sum of the least squares distance between the observed and simulated test incidence by region.

This, however, was not sufficient to give good results in all regions, because transmission was uneven across regions. More populous and densely populated regions tend to experience more cases than smaller and less urban regions. Therefore, in the regional calibration, we computed a vector of multiple distances, one for the sum of the larger/more densely populated regions and one with the sum of the distances for all other regions. In this way, as all components of this vector were minimised in the calibration, resulting in an overall good fit, not only in the regions with most cases. If all regions would instead have been fitted together using one common error measure, the regions with most cases would dominate, because of their largest contribution to the least square sum. In SMC-ABC [12], the parameters are calibrated in subsequent rounds, where the distance between simulated and observed data decreases in each round. We thus used four separate tolerances for the distances, ensuring that they all diminished in each round: hospitalisation incidence in the largest regions, hospitalisation incidence for the rest of the regions, laboratory-confirmed cases in the largest regions, and laboratory-confirmed cases in the rest of the regions. In Norway, infection rates have been consistently high in the densely populated region of Oslo and the neighbouring populous Viken region, and we therefore used a separate tolerance measure for these two regions combined.

In the first round, we started with pre-specified thresholds for the distances, and a prior for all the parameters. Candidate parameter values were drawn from the prior, and we simulated the hospitalisation and positive test time series for each region using these candidate parameters. We accepted the candidate parameters if all the distances were below the pre-specified thresholds. For all subsequent rounds, each threshold was chosen as the 0.80-quantile of the distances produced by the accepted parameters in the previous round. Candidate parameters were then sampled from the set of chosen parameters from the last round using importance weights. To add further exploration, these candidate parameters were perturbed with a multivariate normal distribution with covariance matrix equal to the empirical covariance matrix of the parameters retained in the previous round. We obtained 1000 parameter sets in each round when we calibrated the regional parameters, 200 when we calibrated the national parameters. This pipeline is summarised in Algorithm 1, see also [24]. In the algorithm, the observed hospital incidence data are denoted by $\mathbf{H}$ and the observed positive tests as $\mathbf{T}$. The corresponding simulated hospital incidence is denoted by $\mathbf{H}'$ and the simulated positive tests as $\mathbf{T}'$, and $^{-7}$ denotes a 7-days backwards moving average. Subscript $OV$ denotes that the error is calculated for Oslo and Viken (the largest regions), while $-OV$ means all counties except Oslo and Viken. The algorithm was almost the same for the national calibration, except that we calculated the errors for the simulations aggregated over regions and compared it to the nationally aggregated data. Hence there were only two thresholds for the national algorithm, one for the hospitalisation incidence and one for the test data.

For the test data, we chose to use a 7-days backwards moving average for the covariate $k_t$, and for the observed number of positive tests and the simulated number of positive tests when

calculating the distance. This was done to take into account potential day-of-the-week-effects and allowed for less day-to-day variance than when using the daily data directly.

**Algorithm 1** ABC-SMC

**Initialise:**

Set a starting value $\epsilon_0^{hov}$, $\epsilon_0^{h-ov}$, $\epsilon_0^{tov}$, $\epsilon_0^{t-ov}$ for the tolerances for the hospitalisation data in the largest regions and for the rest of the regions, and test data in the largest regions, and the rest of the regions, respectively, $r = 1$ and $\mathbf{w}^1 = (1, \ldots, 1)$.

set $i = 0$.

**while** $i < 1000$ **do**

 **if** $r = 1$ **then**

 sample parameters $\theta_i$ from the prior $\pi$.

 **else**

 Sample $\boldsymbol{\theta}^p$ from $\boldsymbol{\theta}^{r-1}$ with weights $\mathbf{w}^{r-1}$.

 Propose $\boldsymbol{\theta}_i$ from a normal distribution $N$ centred at $\boldsymbol{\theta}^p$ with variance equal to the empirical variance from the previous round.

 If $\pi(\boldsymbol{\theta}_i) = 0$, sample a new $\boldsymbol{\theta}_p$.

 Run the model with $\boldsymbol{\theta}_i$, providing the simulated number of hospitalised cases $\mathbf{H}'$, and positive tested $\mathbf{T}'$.

 **if** $f^h(\mathbf{H_{ov}}', \mathbf{H_{ov}}) = ||\mathbf{H_{ov}}' - \mathbf{H_{ov}}||_2 < \epsilon_r^{hov}$ and $f^h(\mathbf{H_{-ov}}', \mathbf{H_{-ov}}) = ||\mathbf{H_{-ov}}' - \mathbf{H_{-ov}}||_2 < \epsilon_r^{h-ov}$, $f^t(\mathbf{T_{ov}}', \mathbf{T_{ov}}) = ||\bar{\mathbf{T}}_{\mathbf{ov}}'^7 - \bar{\mathbf{T}}_{\mathbf{ov}}^7||_2 < \epsilon_r^{tov}$ and $f^t(\mathbf{T_{-ov}}', \mathbf{T_{-ov}}) = ||\bar{\mathbf{T}}_{-\mathbf{ov}}'^7 - \bar{\mathbf{T}}_{-\mathbf{ov}}^7||_2 < \epsilon_r^{t-ov}$, **then**

 set $\boldsymbol{\theta}_i^r = \theta_i$, and calculate weights $\mathbf{w}_i^r$ for the parameters, as

$$\mathbf{w}_i^r = \begin{cases} 1, & \text{if } r = 1, \\ \dfrac{\pi(\boldsymbol{\theta}_i)}{\sum_{j=1}^n w_j^{r-1} N(\boldsymbol{\theta}_i | \boldsymbol{\theta}_j^{r-1})}, & \text{otherwise,} \end{cases}$$

 Increment $i = i + 1$.

 **else**

 sample a new $\boldsymbol{\theta}_p$.

normalise the weights as $\mathbf{w}_i^r = \frac{w_i^r}{\sum_j w_j^r}$.

set $\epsilon_{r+1}^{hov}$ to the 80th percentile of $f^h(\mathbf{H_{ov}}', \mathbf{H_{ov}})$ for the accepted parameters.

set $\epsilon_{r+1}^{h-ov}$ to the 80th percentile of $f^h(\mathbf{H_{-ov}}', \mathbf{H_{-ov}})$ for the accepted parameters.

set $\epsilon_{r+1}^{tov}$ to the 80th percentile of $f^t(\mathbf{T_{ov}}', \mathbf{T_{ov}})$ for the accepted parameters.

set $\epsilon_{r+1}^{t-ov}$ to the 80th percentile of $f^t(\mathbf{T_{-ov}}', \mathbf{T_{-ov}})$ for the accepted parameters.

Increment $r = r + 1$.

Concerning the prior distribution, we assumed an independent normal prior for all the variables except the delay between entering the infectious symptomatic or asymptomatic class and a positive test, which was assumed to be a uniform integer between 0 and 4 days. For the reproduction numbers, we used the same prior mean and variance for all the regions. For Norway, the mean was assumed to be 3.7 with variance 0.4 for $R_0$, and 1.0 with variance 0.25 for the other reproduction numbers. For the amplification factor, we assumed a mean of 1.3 and a variance of 0.25. The priors for $R_0$ and the amplification factor were based on early assessments of hospital prevalence data for Norway. For $\pi_0$ and $\pi_1$, we assumed mean 0 and variance 4.0 and $25 \cdot 10^{-9}$, respectively. We truncated the normal distribution at 0 for the reproduction numbers, amplification factor, and $\pi_1$.

**Split-SMC-ABC.** The regional calibration is challenging, as it is a very high-dimensional calibration problem. With the standard SMC-ABC, we were not able to obtain convergence in useful time for our almost 100 parameters. Therefore, we developed a new version of the SMC-ABC to produce reliable and timely results, which we call the Split-SMC-ABC.

The simple idea is to restrict the past reproduction numbers to the already obtained posterior samples, by sampling from the past trajectories and parameters as detailed below. Hence, all the computational time is used to estimate the most recent reproduction numbers, the key parameters for situational awareness. However, parameters that enter the earlier part of the epidemic model are also important, as the more recent reproduction numbers (or equivalently $\beta_t^i$) depend on them. For example, a higher-than-average reproduction number in one period is typically followed by a lower-than-average reproduction number in the following period, to compensate for the high number of cases produced in the first period. In addition, the whole history is necessary for estimating the total cumulative number of infections and hence the immunity of the population. Therefore, since the present parameters depend on the past parameters, we cannot treat the two calibration problems (past and present) as independent. Instead, we ran the past until convergence and used the obtained posterior distributions of the parameters in the past when calibrating the more recent period. Importantly, we needed to handle the transition period between the past period and the more recent period. Here, we exemplify the algorithm with the dates used in the simulations in this paper, but the approach is, of course, general. We focus on the split into two parts, which we call batches. Note that this operation can be repeated multiple times, generating multiple batches, by following the same procedure.

We start with the first batch calibration, obtained by calibrating to all the data up to August 15, 2020. From this calibration, we obtained 1000 posterior samples of transmissibility parameters up to August 1 2020, $\pi_0$, $\pi_1$, $d$, and the amplification factor. We also have the 1000 corresponding epidemic time series of $S^i$, $E_1^i$, $E_2^i$, $I^i$, $I_a^i$, and $R^i$ in each region $i$. The first batch calibration contained a changepoint August 1, 2020 in all regions. We denote this timepoint, August 1, 2020, by $t_{cal}$, which was the first date of the data we wished to calibrate to in the second batch. Hence, all the parameters prior to $t_{cal}$ were restricted to the sampled posteriors of the first batch calibration. We also fixed the time-independent parameters $\pi_0$, $\pi_1$, $d$, and the amplification factor to follow their posterior distribution from the first batch. In the second batch calibration, we calibrated all transmissibility parameters entering the model after $t_{cal}$.

We now focus on the transition between the two batches. For this purpose, the first batch calibration was obtained by calibrating to data up to $t_{cal} + 15$. This is important because of the posterior dependence between consecutive transmissibility parameters. As the simulated hospitalisations and laboratory-confirmed cases on $t_{cal}$ corresponded to simulated transmission events that occurred a certain number of days before, it was necessary to start the simulations a certain number of days before $t_{cal}$. We chose to start the simulations at $t_{cal} - 15$, as 15 days covered approximately 90% of the expected hospitalisations in the simulations in our setting (which are more delayed than the positive tests). Hence, we kept the 1000 calibrated trajectories (along with the corresponding calibrated reproduction numbers) from the first batch calibration, up to the day $t_{cal} - 15$. The simulations in the second batch were started on the exact states ($S^i$, $E_1^i$, $E_2^i$, $I^i$, $I_a^i$, and $R^i$ in each region $i$) that were obtained in the first batch calibration, on the date $t_{cal} - 15$. When continuing one of the 1000 past trajectories from $t_{cal} - 15$ to $t_{cal}$, the corresponding most recent reproduction number was used.

In the second batch, we calibrated only parameters entering the model after $t_{cal}$, using all the available data from $t_{cal}$ onward. In the first round $r = 1$, we first sampled uniformly which of the 1000 trajectories produced in the first batch we should continue (with the corresponding

amplification factor, $\pi_0$, $\pi_1$, $d$, and reproduction number for the first 15 days for that trajectory). We continued to run with different samples of the past trajectories and the new parameters from the second batch, until we had 1000 accepted parameter sets with errors lower than the prespecified thresholds for the test data and the hospital data from $t_{cal}$ until the latest data point.

For all subsequent rounds, $r > 1$, we first sampled which trajectory from the past period we should run. The sampling probabilities $S(p)$ for each past trajectory $p$ were calculated as

$$S(p) = (1/1000 + N_{r-1}(p))/1001,$$

where $N_{r-1}(p)$ is the number of times trajectory $p$ was chosen in the previous round $r − 1$, then normalised to sum to 1. Hence all past trajectories had non-zero probability of being chosen. The probability of selecting a specific trajectory increased linearly with how frequently it was accepted in the previous round. We chose to do this as it could be that certain past trajectories were more likely, given the current data. We divided by 1001 to ensure that the probabilities sum to 1. Then we sampled from the 1000 accepted parameter sets for the second batch from the accepted parameter sets of the previous round in the usual way, as described in Algorithm 1.

Using the split method, we lose some of the dependence structure between the past and the most recent reproduction numbers. We limited this effect by letting the last reproduction number in the first batch be informed by future data in a period of 15 days after $t_{cal}$. As mentioned, this had good motivations and worked well in practice in the case of Norway. Other lengths of the overlap should be tested in other cases. From a theoretical point of view, it is difficult to quantify how much stochastic dependence is lost in this way.

For Norway, we used two additional temporal split points in the calibration, resulting in four different batches. One before August 1, 2020, one between August 1, 2020 and November 5, 2020, one between November 5, 2020 and January 4, 2021, and one after January 4, 2021.

A demonstration of the calibration performance in a simple setting is provided in the S1 Text Section S2.11. Note that we have yet to investigate the theoretical convergence properties of the method.

**Regionally separate prior calibrations.** We noticed that this calibration method, when ran for a reasonable number of rounds, produced parameter estimates which typically led to good fits for the most populated regions and/or regions with more cases. The fit was not particularly good for the regions with fewer cases.

As already mentioned, we introduced a vector of calibration distances in the SMC-ABC, with one distance measure for the cases in large regions and one combining the rest of the regions, by summing their distance measures. To ensure a good fit in all regions, we would have preferred to introduce one distance measure for each region. However, it is not feasible to work with such a high-dimensional error measure, as the probability of sampling a parameter vector that improves the fit in all regions simultaneously would be very low. Hence, such a high dimensional error measure would result in slow convergence. Therefore, we propose a two-step version of the split-SMC-ABC for the final batch.

We first performed one separate calibration for each region. This was done using the same disease spread model and setup as previously described, but assuming a common, national transmissibility for all the other regions, except the one of interest. We then took the separately calibrated transmissibility parameters for each region and used their posterior distribution as prior distributions in the regional calibration which simultaneously calibrated the transmissibility for all the regions. This last step was necessary to learn the spatial correlations, as the cases in the different regions are dependent due to the mobility between

the regions. The algorithm for the separate calibrations for each region was the same as the algorithm provided in Algorithm 1, except that the error was separated into the region of interest and the rest of the regions, instead of the largest regions and the rest of the regions.

## Results

### Regional reproduction numbers

The estimated regional effective reproduction numbers are provided in Table 1 for the early part of the Norwegian pandemic, for the rest see S1 Text, Section S2.3 and Tables E-G. We observed differences between regions, both in estimates and uncertainty. The estimates were higher and most certain for the counties Oslo and Viken, with the largest population and with most cases.

The lockdown effect on reproduction numbers was significantly different between regions, with reductions ranging from 69% to 94% compared to before lockdown. After reopening schools and kindergartens, the reduction diminished to between 77% and 88% of the pre-lockdown setting. In Oslo, the reduction due to the lockdown was estimated to 89% (95% CI 84%-93%), and later 80% (75%-84%) when lockdown was eased.

**Table 1. Estimated reproduction numbers.**

| Region | $R_0$ until 03/14 | $R_1$ 03/15–04/19 | $R_2$ 04/20–07/31* |
|---|---|---|---|
| Viken | 3.92 (2.95–4.72) | 0.27 (0.09–0.47) | 0.84 (0.74–0.96) |
| Oslo | 5.11 (4.44–5.85) | 0.54 (0.41–0.71) | 1.04 (0.95–1.13) |
| Vestland | 3.28 (2.08–4.48) | 0.33 (0.05–0.55) | 0.51 (0.11–0.91) |
| Rogaland | 3.48 (2.28–4.43) | 0.2 (0.03–0.4) | 0.68 (0.1–1.15) |
| Trøndelag | 3.64 (1.85–5.53) | 0.62 (0.31–0.98) | 0.67 (0.32–0.91) |
| Vestfold og Telemark | 3.13 (1.53–4.84) | 0.25 (0.02–0.57) | 0.39 (0.07–0.69) |
| Innlandet | 3.73 (1.95–5.39) | 0.54 (0.2–0.84) | 0.49 (0.07–0.8) |
| Agder | 2.99 (1.68–4.04) | 0.32 (0.06–0.56) | 0.51 (0.08–0.96) |
| Møre og Romsdal | 3.41 (1.26–5.59) | 1.06 (0.84–1.29) | 0.42 (0.04–0.94) |
| Troms og Finnmark | 3.26 (1.85–4.47) | 0.25 (0.03–0.65) | 0.75 (0.17–1.27) |
| Nordland | 3.89 (1.44–6.24) | 0.38 (0.04–0.81) | 0.63 (0.17–1.08) |
| Region | Reduction $R_1$ | Reduction $R_2$ | Population (density [km$^{-2}$]) |
| Viken | 0.93 (0.84–0.98) | 0.79 (0.67–0.84) | 1 241 165 (55) |
| Oslo | 0.89 (0.84–0.93) | 0.8 (0.75–0.84) | 693 494 (1 628) |
| Vestland | 0.9 (0.74–0.99) | 0.84 (0.56–0.98) | 636 531 (20) |
| Rogaland | 0.94 (0.82–0.99) | 0.80 (0.5–0.98) | 479 892 (26) |
| Trøndelag | 0.83 (0.47–0.94) | 0.82 (0.51–0.94) | 468 702 (12) |
| Vestfold og Telemark | 0.92 (0.63–0.99) | 0.88 (0.55–0.99) | 419 396 (419) |
| Innlandet | 0.86 (0.57–0.96) | 0.87 (0.59–0.99) | 371 385 (8) |
| Agder | 0.89 (0.67–0.99) | 0.83 (0.43–0.98) | 307 231 (21) |
| Møre og Romsdal | 0.69 ((-0.02)-0.85) | 0.88 (0.25–0.99) | 265 238 (265) |
| Troms og Finnmark | 0.92 (0.65–0.99) | 0.77 (0.31–0.96) | 243 311 (243) |
| Nordland | 0.9 (0.44–0.99) | 0.84 (0.25–0.97) | 241 235 (7) |

Top: Estimated regional effective reproduction numbers (mean and 95% CI) in the early period, 2020. Bottom: Estimated relative reduction of transmissibility (mean and 95% CI) at lockdown start and stop, population size and density [25].

*Oslo, Viken, Nordland, Innlandet, Vestfold og Telemark og Vestland 04/20–07/24.

Illustrative examples of the fit to the daily hospitalisation and laboratory-confirmed case incidence for some counties is provided in Figs 3 and 4, respectively. The uncertainty varied and was naturally larger for counties with fewer cases (e.g., Nordland).

In S1 Text Section S2.9, we compare our estimated reproduction numbers for Oslo and Viken to estimates from a non-parametric estimation method implemented in the R-package EpiNow2 [34].

## The national picture

To present a national picture, we calibrated the model assuming the same transmissibility in all regions (Table 2). In Norway, the estimated national reduction in transmissibility after the lockdown in March 2020 was 85% (78%, 89%), to a reproduction number significantly below 1. After reopening schools, we estimated a reduction of 79% (65%, 91%) compared to before interventions. During late spring 2020, many restrictions were lifted, but the estimated mean/ median effective reproduction number stayed below 1 until August 1, 2020, when the borders, universities and schools reopened. During autumn 2020, the estimated national reproduction number increased. Many restrictions were again implemented on November 5, 2020, and we estimated a reduction of 75% (69%, 80%) compared to $R_0$. For the restrictions on January 4, 2021, we estimated a reduction of 81% (76%, 85%) compared to $R_0$. On March 2, 2021, Oslo implemented several restrictions. Nationally, we estimated an effect of these interventions of 66% (57%, 73%) compared to $R_0$.

In S1 Text, Section S2.6, we further discuss and present changes in mobility data associated with the intervention policies.

As part of the calibration process, we also estimated the proportion of imported cases that went undetected. Whenever the disease level is low, importation of the virus into the regions is decisive. In Norway, many of the early imported cases were identified, as testing of everyone returning from known risk areas was recommended [35]. We estimated that only 49% (33%, 84%) of the imported cases were notified and registered.

## Regional predictions

Regional predictions of hospitalisations have been essential for preparedness and capacity planning in Norway. We produced three-weeks-ahead predictions. Importantly, our predictions assumed no changes in the transmissibility in the short-term, hence showing what would happen if policies and behaviour remained unchanged. However, as the predictions were used to inform decision-making by the Norwegian government, publicly available and communicated by the media, they often triggered new policies and behavioural changes, which in turn influenced reproduction numbers.

As an example, in Fig 3, we show three-weeks-ahead predictions of hospitalisations, using data until April 1, 2021. As additional interventions were implemented in Norway in late March 2021, our predictions overestimated the actual data in the county Viken and nationally. This was due to the natural time gap between transmission and laboratory-confirmed test and/ or hospitalisation: the changes were not yet visible in the data used for calibration. In S1 Text, Figs G-Z, we present predictions for other dates, where the predictions were better because intervention policies remained unchanged during the prediction period.

When the main interest is predicting hospitalisations, it is unclear whether it is best to use both test and hospitalisation data or only the latter when calibrating. On one side, using all available data should reduce uncertainty, and test data contain more information about recent changes because of the shorter delay. But if the two data sources are not coherent, parameter estimates are more uncertain and possibly biased, so in turn, also the hospital predictions may

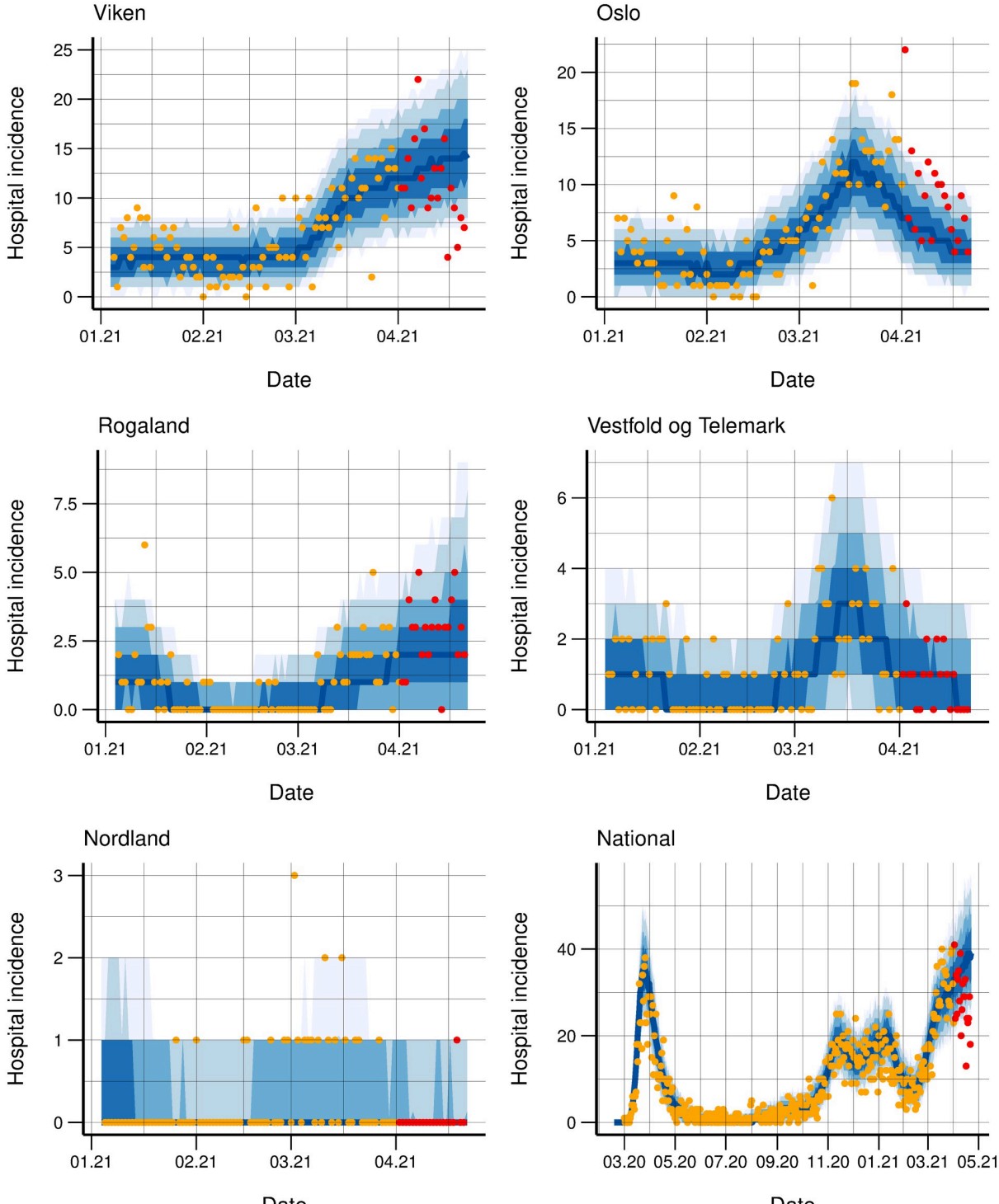

**Fig 3. Hospital incidence fit.** Observed daily (orange dots) and simulated (blue bands) hospitalisation incidence January to April, 2021 for five counties and nationally March 2020 to April 2021. 3-week-ahead predictions (blue bands) from April 1, 2021 are included together with actual data (red dots), which are not used in the calibration. The bands correspond to 50, 75, 90 and 95% credibility intervals.

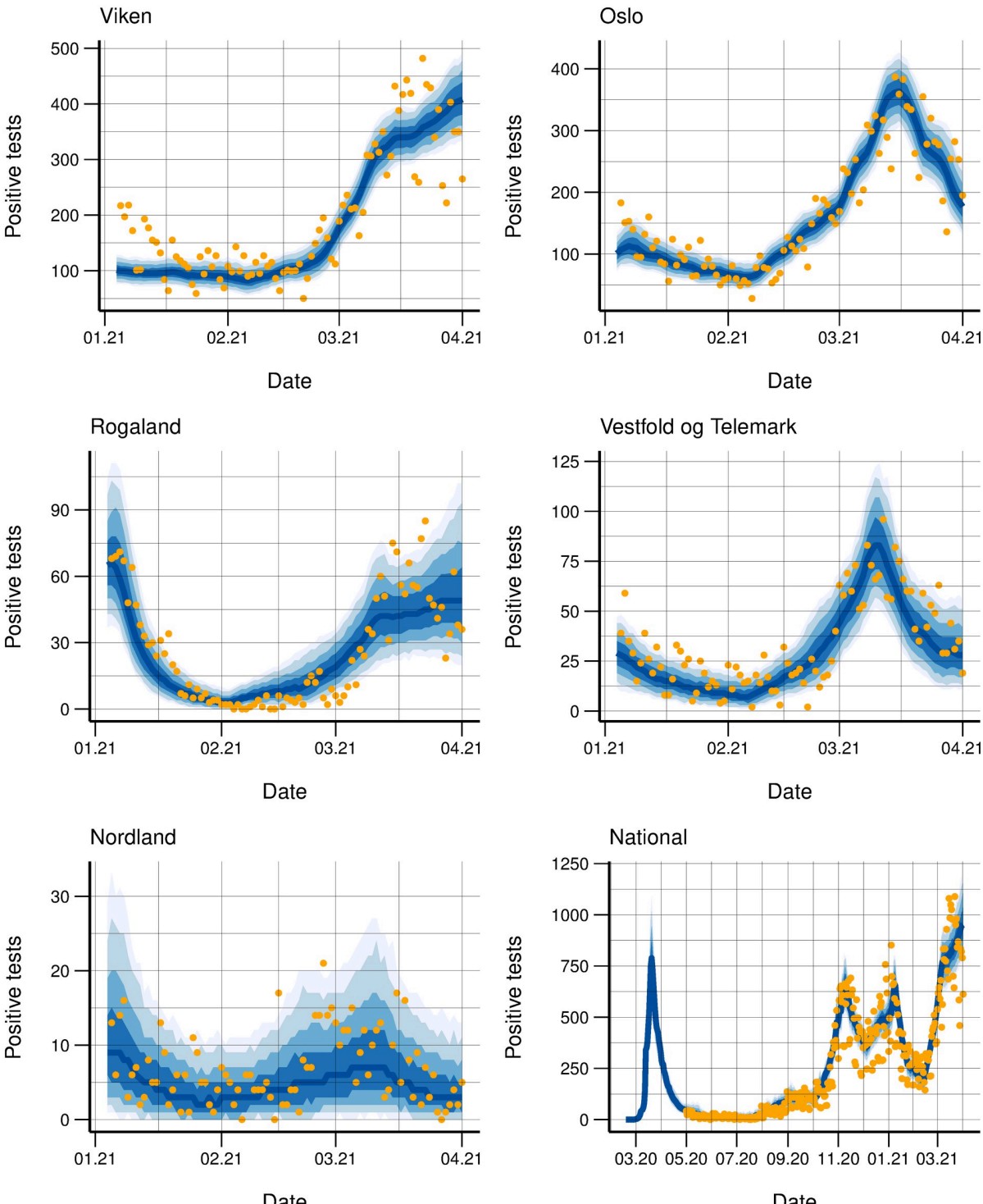

**Fig 4. Test data fit.** Observed daily (orange dots) and simulated (blue bands) laboratory-confirmed cases January to April, 2021 for five counties and nationally March 2020 to April 2021. The bands correspond to 50, 75, 90 and 95% credibility intervals.

**Table 2. Estimated national effective reproduction numbers (mean and 95% CI).** Dates are provided as mm/dd/yyyy.

| R | Period |
|---|---|
| 3.24 (2.47–3.98) | 02/17/2020–03/14/2020 |
| 0.49 (0.41–0.58) | 03/15/2020–04/19/2020 |
| 0.66 (0.29–1.1) | 04/20/2020–05/10/2020 |
| 0.65 (0.12–1.09) | 05/11/2020–06/30/2020 |
| 0.95 (0.14–1.63) | 07/01/2020–07/31/2020 |
| 1.09 (0.73–1.36) | 08/01/2020–08/31/2020 |
| 0.9 (0.73–1.1) | 09/01/2020–09/30/2020 |
| 1.28 (1.09–1.47) | 10/01/2020–10/25/2020 |
| 1.23 (1.04–1.5) | 10/26/2020–11/04/2020 |
| 0.81 (0.75–0.87) | 11/05/2020–11/30/2020 |
| 1.06 (1.02–1.12) | 12/01/2020–01/03/2021 |
| 0.6 (0.5–0.71) | 01/04/2021–01/21/2021 |
| 0.8 (0.65–0.93) | 01/22/2021–02/07/2021 |
| 1.5 (1.39–1.64) | 02/08/2021–03/01/2021 |
| 1.08 (1.01–1.14) | 03/02/2021–04/01/2021 |

be substantially biased. We compared hospital predictions when calibrating to both test and hospital data, to when calibrating only to the latter. Results showed that point predictions were more accurate when only calibrating to the hospital data, see S1 Text, Figs K-P. The predicted hospitalisations were systematically underestimated when the test data were included. However, the prediction uncertainty was larger when using only the hospitalisation data.

## Improved predictions using regional transmissibilities

We investigated the predictive performance of the proposed model with regionally varying reproduction numbers for Norway by comparing it to the model using a nationally constant reproduction number. In addition, we compared it to a simple regional baseline calibration model that predicts new data will be equal to data from the last two weeks without any trend. The model is described further in S1 Text, Section S1.4. We predicted the weekly hospitalisation incidence and the number of new confirmed symptomatic cases region-wise for three weeks ahead following April 1, 2021, March 1, 2021, November 1, 2020, October 1, 2020, and September 1, 2020. As the calibration is computationally time consuming, we restricted the comparison to five periods, selected to cover both relatively stable and changeable parts of the epidemic. We calibrated using data up to the prediction date. To quantify the quality of predictions and compare models, we calculated a multivariate energy-score for the region-wise predictions and a continuous ranked probability score (CRPS) for the aggregated national predictions [36], using the R-package scoringRules [37]. Proper scoring rules allowed us to study the quality of probabilistic forecasts by simultaneously considering the overlap between the predictions and observations and the width of the forecast distribution. Lower scores indicated better predictive performance. Scores of predictions from different models can be compared to conclude which model is the preferred model. However, the scoring rules do not provide an interpretable quantification of how well the predictions performed in absolute terms. We also present the percentage of weeks per region when the observed hospital incidence was within the 95% prediction interval (PI).

Results, averaged over the five periods, are shown in Fig 5, see also S1 Text, Table D. The model with regional reproduction numbers clearly outperformed the national model on

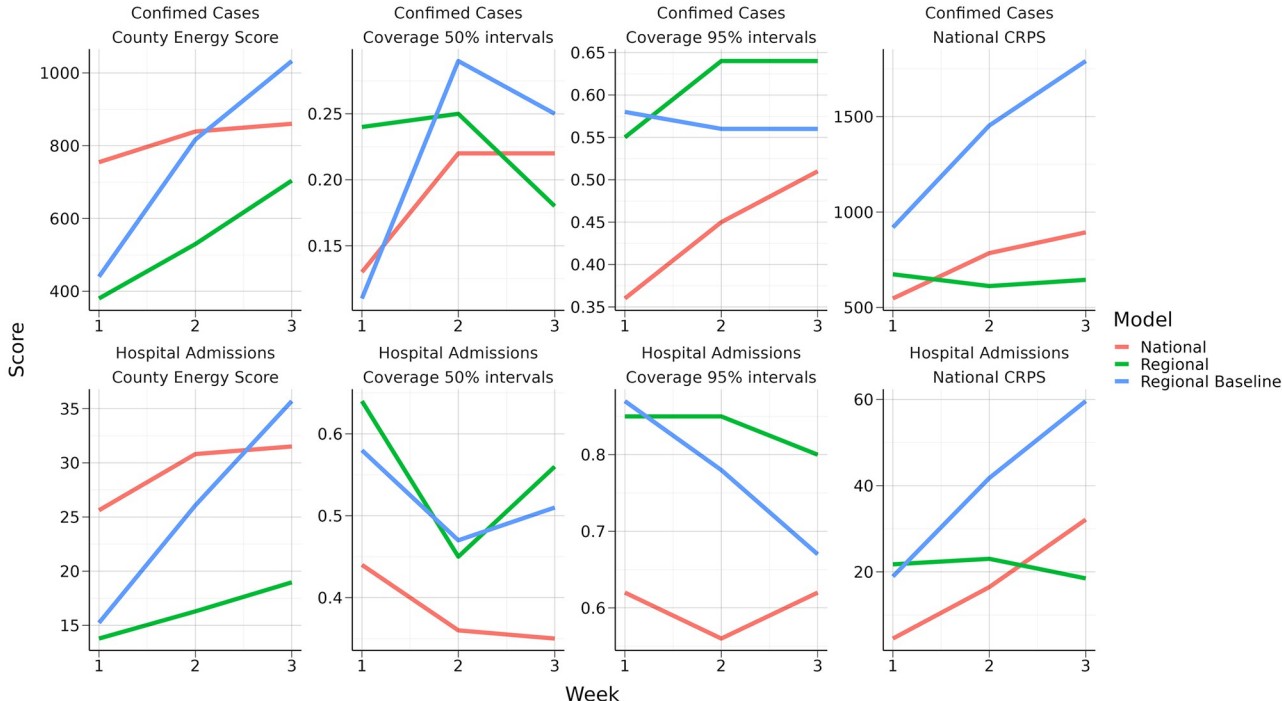

**Fig 5. Performance scores.** Average county energy score, national CRPS and county level coverage of 50 and 95% PI for the regional and national model.

predicting hospital incidence and confirmed cases at the county level, shown by the regional energy scores. The national model performed slightly better on the aggregated national CRPS score for shorter predictions of 1 week, but the regional model performed better at national level for predictions two and three weeks ahead. The regional model also exceeded the simple regional baseline model. The coverage of the regional 50% and 95% PI was good for the hospital admissions, but low for the test data. This was probably since we used a 7-days moving average in the calibration of the test data, to avoid estimating additional day-of-the-week effects. In Norway, short term predictions of hospital admissions have been an important public health tool, while predictions of the number of confirmed cases have mainly been used for model checking and validation.

## Discussion

We present a new calibration framework and regional model for situational awareness and surveillance of a pandemic. During a pandemic like the COVID-19 pandemic, timely information about the local effects of most recent interventions is crucial for operational policy decisions. We tested our model in the case of Norway where many essential aspects are present: incompletely observed importation; time-varying testing capacity and test-targeting strategies; focus on regional capacity limits of the health system; nation-wide and local interventions; uncertain changes in population behaviour; spatial heterogeneity of the epidemic with clusters in the largest cities.

Regarding the early phase of the pandemic in Norway, we estimated a larger lockdown effect in the most populated regions which also had the most cases, compared to the the national average reduction. Similar results were found for London (81% [38]) compared to the

whole UK. Note that the estimated reproduction numbers in that period are on the high side of the estimated reproduction numbers for other countries [39], which could be due to for example an underestimation of importation, underestimated hospitalisation risks, longer assumed generation time, or due to mobility, as illustrated in S1 Text, Section S2.8.

Nationally, we estimated a reduction of 85% in transmissibility due to the first lockdown, similar to the UK (estimated reduction 75% [38]), Germany (79% [40]) and all Europe (81% (range 75% to 87%) [41]).

Though we found that the early interventions successfully contained the epidemic, we are not able to discern how the various measures contributed. Interestingly, we found that the reopening of kindergartens and schools did not increase the reproduction number significantly in Norway. Note however that a small increase in the reproduction number may lead to a large increase in the number of cases in a setting close to the epidemic threshold. Similar findings have been reported in China, where the increase in intra-city movement after the lockdown was negatively correlated with transmissibility [42]. We also found that the transmission rate did not grow significantly when mobility was almost back to normal. Hence, mobility appeared not to drive infectious contacts.

There was a persistent difference in the course of the epidemic in the various counties, which the regional model accounted for through significantly different estimates of local effective reproduction numbers. The regional model performed better at predicting hospitalisations than a model assuming the same transmissibility nationally, and a zeroth-order regional model forecasting assuming static case counts. Hence, estimating regional reproduction numbers is possible and can improve local hospital planning. We selected the changepoints manually, guided by regionally differentiated interventions and levels of infection. The model performance depends on their placements. Failure to add a changepoint when there is a significant change in transmission makes the model perform badly because more parameters are needed to cope with non-stationarity. To this end, methods for data-driven changepoint detection [40] can be useful and should be tested in our model in the future. The three-weeks-ahead predictions could be improved by incorporating estimated effects of planned interventions during this period. For this purpose, data on the effect of different restrictions (both isolated and in combination) on regional transmissibility are necessary. Such data are not yet available and is a theme for further research.

Commenting on the modelling approach, we document how to use both laboratory-confirmed cases and the hospitalisation data to inform our regional model and show the importance of the integration of both data sources. Hospital incidence data are reliable, but suffer from a time delay. The laboratory-confirmed case data are less delayed and contain information on the most recent days; however, repeated changes in testing criteria, capacity, and technologies, make test data difficult to de-bias, as needed for inference. Inconsistency is known to be challenging [43] and it results in compromising parameter estimates between the data sources. While this was useful to make estimation of reproduction numbers more precise, we showed that when predicting hospital occupancy, calibrating to only hospital data resulted in less bias, but considerably higher variance.

Inference for many parameters is necessary in a regional model and has been an insurmountable challenge for useful-computational-time operations. Our new calibration method consists of series of calibrations each with fewer parameters, by splitting the calibration data into different periods and moving chronologically. Once the past parameters have been estimated, they are restricted to the samples from the posterior distribution so that one can use all the computational time in operation to estimate the most recent reproduction numbers. However, despite the split approach, the approach is still computationally time consuming due to the increasing difficulty of obtaining acceptable parameter estimates in each round of the calibration.

A key feature of our model is the lack of age structure in the disease dynamics, which would require many more parameters. Instead, we have suggested a data-driven approach, using the age distribution of the positive test cases to describe the age-profile of transmission. In this way we can account for behavioural changes in the population, avoiding a priori assumptions about mixing patterns between age groups.

We did not include additional compartments for vaccinated individuals in the metapopulation model. That would have required multiple assumptions about the time-varying effects of the different vaccines. Instead, we estimate the effective reproduction number after vaccination directly from the data. Hence, the estimated effective reproduction number automatically captures the population which is immune due to vaccines. In Norway, vaccination was prioritised to the elderly and then gradually expanded from older to younger populations as more vaccines were available. Since we use the age profile of the test data to calculate hospitalisation risks, the effect of the vaccination is visible as a decay in hospitalisation risks for the vaccinated age groups and in this way enter the model. By April 1, 2021, approximately 5.4% and 7.5% of the Norwegian population had received their second and first vaccination dose, respectively.

The model assumes complete and lasting protection from infection. With the emergence of new variants, particularly the Omicron variant, reinfections are more frequent. However, extending the model framework to include partial protection or waning immunity is relatively straightforward to implement, for example by moving individuals from the recovered compartment back to the susceptible compartment or introducing additional vaccination compartments, but it of course requires additional assumptions.

We used a single nationwide detection probability for the laboratory-confirmed cases, reflecting the national testing criteria in Norway. The estimated probabilities were in the range 55–70%. These proportions are high compared to values of 10–40% reported from Italy and France. Regularly collected seroprevalence data could provide crucial information about population incidence of SARS-CoV-2 infection in Norway; however, unfortunately such data was not available. A single large-scale Norwegian seroprevalence study suggests that around 9 in 10 cases were detected [44].

Testing practice and thus detection probabilities may differ between regions. It is possible to estimate separate detection probabilities per region, but with a significant computational cost. In addition, testing criteria have varied in time, independently of the number of tests performed, affecting the consistency between test and hospital data.

In the case of Norway, we opted for a county-level description in the regional model. The choice of spatial scale is a trade-off between accuracy/detail and computational time. The computational time for the infectious disease model scales quadratically in the number of regions, and hence increasing the number of regions is costly. In addition, if the population size in a region is too small, there might be too little signal in the data to obtain useful estimates. If the region is too large, it might be unreasonable to assume homogeneity within the region.

The metapopulation model is appropriate for incorporating mobility estimates on an aggregated level, like in our setting. It allows us to build a spatial prediction model, which can provide local estimates of infection level. Intra-regional details are not included in the model. This means that the model is for example not the most appropriate to estimate the effect of different isolated interventions, like closing of schools. For this, the more detailed, data-hungry, and computationally demanding agent-based models should be used.

We assume that the mobile phone mobility data are representative of the whole population. In reality, there are likely sources of bias in the mobility data, for example, age bias (very young and old individuals less often possess a mobile phone and are thus not well captured). However, in the case of Norway, Telenor Norway has a substantial market share, well distributed

over all age groups. While there are published SARS-CoV-2 models informed by historical mobility data, e.g. [45] using commuting data from 2011, to the best of our knowledge, our modelling approach is first to employ real-time national mobility data with fine geographical resolution.

The SEIR model operates with six-hour intervals where the latest recording during the final two hours is used to track location. Of course, our construction misses movements that happen during the four non-monitored hours. We have chosen six hours as the temporal resolution to capture both commuting and long-distance trips. With a shorter time interval, we would better capture short-duration/short-distance movements at the cost of losing information on long-distance/long-duration travel. Vice versa, we would measure long-distance/long-duration travel better with a longer time window. We believe that the short-duration movements are less critical for the epidemic spread. Moreover, the smaller the time resolution, the more noise and randomness, as fewer individuals move. In addition, a more refined time scale is also computationally more expensive. The reason why we only use a single location registration every six hours is computational: the full history of visited cell towers is extremely vast, and it is computationally hard to process such complete data. However, this should not be critical to the workings of the model since during the night, movements are limited, and trips of short duration are less critical for the spread of the epidemic.

We used mobile phone data lacking information about the home residency of users. Using simulation, we have explored the effect of applying different rules for selecting individuals to transfer based on their home municipality. We measured the effects on spatiotemporal dynamics by determining the timing and size of the peak of the epidemics when using different rules or with the regularisation method. We show that our choice of prioritising visitors (from target municipality) before hosts (from donor municipality) and, lastly, residents of separate localities broadly gives similar spatiotemporal dynamics as the other rules we tested. Moreover, we document that mobility data generated via our regularisation method produce similar spatiotemporal dynamics as the original origin-destination mobility matrices when applied with the mobility rules used in this paper, supporting the broader use of mobility data in real-time settings.

Our predictions and estimates rely on several parameters which are uncertain. There are uncertainties related to the time from symptom onset to hospitalisation, the age-specific hospitalisation risks, and the natural history parameters of COVID-19. We have based our estimates on local Norwegian data where available, and otherwise relied on international studies. Seroprevalence surveys can provide an understanding of the population-level incidence of COVID-19. Due to a lack of such data from Norway, the risk of hospitalisation is uncertain. Repeated nationwide seroprevalence surveys are essential to improve the accuracy of the model results. Our model is intrinsically dynamic, in the sense that we update the parameters as new data points arrive daily, and when more information becomes available. This also implies that the results have changed during the pandemic, sometimes significantly.

Finally, in this study we have worked with historic data where the case counts are known. This situation is different from doing real-time surveillance and forecasting, where recent data are subject to reporting delays. In operation, we therefore additionally estimate reporting delays. This is done by modelling the progressive correction of data, estimated using a binomial model of the proportion of cases that have been reported the last one, two, three, and four days, corresponding to the maximally observed reporting delay.

In conclusion, regional differences are a key trait of the COVID-19 pandemic. We propose a stochastic regional metapopulation model for situational awareness including real-time mobility data, calibrated to combined daily counts of cases and admissions using a novel Split-

ABC-SMC technique. We provide estimates of county-specific and national reproduction numbers for Norway, and three-weekly projections of regional hospital and ICU beds, thereby documenting our modelling pipeline supporting the Norwegian health authorities and government since the early phase of the COVID-19 pandemic. Our approach is rather complex, focussing on geographical aspects at the expense of adopting a simple transmission model within each unit. We show that the model provides better projections compared to selected reference models. However, other studies, including adaptations to other countries, and additional data, are warranted to validate the approach further.

## Supporting information

**S1 Text. Supporting information appendix for: A real-time regional model for COVID-19: Probabilistic situational awareness and forecasting.** Additional supplementary results and methods.
(PDF)

## Acknowledgments

The authors are grateful to Geir Storvik, Magne Aldrin, Gry Marysol Grøneng and Pia Karoline Abel-Zur Wiesch Genannt Hülshoff for discussions on models during the COVID-19 pandemic in Norway. We are grateful to Norwegian Institute of Public Health, Beredskapsregisteret for COVID-19 for gathering the Norwegian COVID-19 data, making it available for use.

## Author Contributions

**Conceptualization:** Solveig Engebretsen, Alfonso Diz-Lois Palomares, Gunnar Rø, Kenth Engø-Monsen, Francesco Di Ruscio, Arnoldo Frigessi, Birgitte Freiesleben de Blasio.

**Data curation:** Solveig Engebretsen, Alfonso Diz-Lois Palomares, Gunnar Rø, Anja Bråthen Kristoffersen, Jonas Christoffer Lindstrøm, Kenth Engø-Monsen, Meghana Kamineni, Louis Yat Hin Chan, Ørjan Dale, Jørgen Eriksson Midtbø, Kristian Lindalen Stenerud, Richard White.

**Formal analysis:** Solveig Engebretsen, Alfonso Diz-Lois Palomares, Gunnar Rø, Anja Bråthen Kristoffersen, Jonas Christoffer Lindstrøm, Kenth Engø-Monsen, Meghana Kamineni, Louis Yat Hin Chan, Ørjan Dale, Jørgen Eriksson Midtbø, Kristian Lindalen Stenerud, Francesco Di Ruscio, Arnoldo Frigessi, Birgitte Freiesleben de Blasio.

**Funding acquisition:** Arnoldo Frigessi, Birgitte Freiesleben de Blasio.

**Investigation:** Solveig Engebretsen, Alfonso Diz-Lois Palomares, Gunnar Rø, Anja Bråthen Kristoffersen, Jonas Christoffer Lindstrøm, Kenth Engø-Monsen, Meghana Kamineni, Louis Yat Hin Chan, Francesco Di Ruscio, Arnoldo Frigessi, Birgitte Freiesleben de Blasio.

**Methodology:** Solveig Engebretsen, Alfonso Diz-Lois Palomares, Gunnar Rø, Anja Bråthen Kristoffersen, Kenth Engø-Monsen, Meghana Kamineni, Louis Yat Hin Chan, Ørjan Dale, Jørgen Eriksson Midtbø, Kristian Lindalen Stenerud, Francesco Di Ruscio, Richard White, Arnoldo Frigessi, Birgitte Freiesleben de Blasio.

**Project administration:** Birgitte Freiesleben de Blasio.

**Resources:** Birgitte Freiesleben de Blasio.

**Software:** Solveig Engebretsen, Alfonso Diz-Lois Palomares, Gunnar Rø, Anja Bråthen Kristoffersen, Jonas Christoffer Lindstrøm, Kenth Engø-Monsen, Meghana Kamineni, Louis Yat Hin Chan, Ørjan Dale, Jørgen Eriksson Midtbø, Kristian Lindalen Stenerud, Francesco Di Ruscio, Richard White.

**Supervision:** Solveig Engebretsen, Arnoldo Frigessi, Birgitte Freiesleben de Blasio.

**Validation:** Solveig Engebretsen, Alfonso Diz-Lois Palomares, Gunnar Rø, Anja Bråthen Kristoffersen, Kenth Engø-Monsen, Meghana Kamineni.

**Visualization:** Solveig Engebretsen, Alfonso Diz-Lois Palomares, Gunnar Rø, Anja Bråthen Kristoffersen, Jonas Christoffer Lindstrøm, Kenth Engø-Monsen, Meghana Kamineni, Louis Yat Hin Chan, Arnoldo Frigessi, Birgitte Freiesleben de Blasio.

**Writing – original draft:** Solveig Engebretsen, Alfonso Diz-Lois Palomares, Gunnar Rø, Anja Bråthen Kristoffersen, Jonas Christoffer Lindstrøm, Kenth Engø-Monsen, Meghana Kamineni, Louis Yat Hin Chan, Arnoldo Frigessi, Birgitte Freiesleben de Blasio.

**Writing – review & editing:** Solveig Engebretsen, Alfonso Diz-Lois Palomares, Gunnar Rø, Anja Bråthen Kristoffersen, Jonas Christoffer Lindstrøm, Kenth Engø-Monsen, Meghana Kamineni, Louis Yat Hin Chan, Ørjan Dale, Jørgen Eriksson Midtbø, Kristian Lindalen Stenerud, Francesco Di Ruscio, Richard White, Arnoldo Frigessi, Birgitte Freiesleben de Blasio.

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
