## [Decision Letter · Decision Letter 0]

5 May 2022

Dear Dr de Blasio,

Thank you very much for submitting your manuscript "A real-time regional model for COVID-19: probabilistic situational awareness and forecasting" for consideration at PLOS Computational Biology.

As with all papers reviewed by the journal, your manuscript was reviewed by members of the editorial board and by several independent reviewers. In light of the reviews (below this email), we would like to invite the resubmission of a significantly-revised version that takes into account the reviewers' comments.

We cannot make any decision about publication until we have seen the revised manuscript and your response to the reviewers' comments. Your revised manuscript is also likely to be sent to reviewers for further evaluation.

Sincerely,

Alex Perkins

Associate Editor

PLOS Computational Biology

Virginia Pitzer

Deputy Editor-in-Chief

PLOS Computational Biology

Reviewer's Responses to Questions

**Comments to the Authors:**

Reviewer #1: The authors present a metapopulation model of SARS-CoV-2 transmission in Norway including the effect of human mobility from mobile phone data, informed with highly detailed information from individual-based surveillance data and calibrated to aggregated data on hospital incidence and notified cases. The model was actually used to support national decision making throughout the COVID-19 epidemic and in this paper it is applied retrospectively to the whole epidemic history up to April 1, 2021. The model is able to reproduce correctly the epidemic curves of hospitalized and confirmed cases by region, and to predict with good accuracy observed data up to three weeks after the end of the calibration period. The manuscript also presents an innovative and well thought algorithm for calibrating the many parameters of the model (mainly the changepoints in regional transmissibility following the initiation and termination of interventions). All in all, the manuscript is clearly written, the methods have been explained with detail and transparency, and the amount of work for its development and calibration is to be commended. Below are some of my comments which I hope can improve this interesting work.

* Parameters for the transmission dynamics (Table S3) were assumed from a preprint coming from the early phase of the pandemic when most of these values were still poorly quantified; furthermore, the purpose of that reference was to demonstrate the effectiveness of digital contact tracing via a theoretical model rather than to realistically describe observed epidemic dynamics. Some of the adopted parameters are not very credible with what is known now about COVID: in particular, the infectiousness of asymptomatic individuals being 1/10 of symptomatic individuals or the 40% probability of being asymptomatic are both very low. I think the authors should verify the impact of these assumptions with sensitivity analyses; I am not asking to recalibrate the model if this is a computationally intensive task, but to demonstrate in the supplementary material how the behavior of the model would change if more realistic parameter values were used. I also recommend finding support for the other parameters used (lengths of incubation/presymptomatic/infectious periods) from relevant epidemiological studies.

* The definition of the model for the detection probability (lines 311+) is quite hard to assess. If I understand correctly, the authors assume that the fraction of detected cases depends logistically on the number of tests performed; however, this does not take into account the fact that the scale-up in the number of tests could not always keep up with the rise in the number of infections (as shown by the positivity test ratio usually going up with new epidemic waves). As a consequence, a detection rate of 55-70% (Figure 2) seems extremely high even for the high public health standards of Norway; this ratio has been estimated at about 10% for Italy and France (Marziano et al, PNAS 2021, Pullano et al, Nature, 2021) in the early phase of the epidemic and to not exceed 40% throughout 2020, even when the incidence was low and the logistic of testing had been fine-tuned.

* The estimated values of the basic reproduction numbers (Table 1) seem very high compared to literature values of 2.5-3. The higher effectiveness found for the lockdown in Norway, compared to estimates from other countries, may derive from an overestimation of the R0. Furthermore, where regional analysis have been conducted, limited variability in both the basic reproduction number and the effectiveness of lockdown was observed (e.g., [Guzzetta et al., EID, 2021]), whereas in this analysis the variability seems significant. Did the authors try to compute the net reproduction number from the epidemic curves at the beginning of the epidemic [Cori et al, AJE 2013], to see how they compare with model-derived estimates? Can the authors discuss their results and provide hypothesis on why would the R0 in Norway be almost 25% higher than in most other countries and the one in Oslo as high as 5.1?

* The conclusion at line 673-674 that "the reopening of kindergartens and schools did not increase transmissibility significantly" needs to be more explicitly contextualized to the lockdown effectiveness in Norway, otherwise it risks to be misinterpreted as a proof of the lack of importance of kindergartens and schools for transmission. Even in the author's analysis, the reopening contributed with an increase of 0.17 (i.e. 35%), in the reproduction number after April 20, 2020 (Table 2); this increase may not be important when starting from a situation of transmission significantly below threshold, but can cause large epidemics if the transmissibility before reopening is close to the epidemic threshold.

* lines 726-728: please note that at least another attempt was previously published early on in the pandemic (Gatto et al., PNAS 2020), so consider revising this sentence.

* It's not clear to me how much regularized mobility matrix (lines 133-136) deviated from the observed matrix and how the artifacts introduced via regularization may impact model outcomes. Can the authors comment on that and possibly add some of the reasoning to the Supplementary information?

* The manuscript is very verbose; A large part of methodological explanations should be moved to the Supplementary information.

Reviewer #2: The paper presents an adapted algorithm for implementing SMC-ABC methods for fitting a discrete-time stochastic epidemic model stratified by regions. The model integrates complex data on movements between regions based on mobile phone data to capture the spread of transmission between regions and to improve the ability to make short-term predictions on the epidemic spread. The model has clearly been in use throughout the COVID pandemic and has been guiding/informing public policy throughout. It's important that such models achieve good visibility with high impact publication so that there is public trust and transparency in the work being done to inform government.

My main concerns regarding the paper are that model has lots of facets and interesting features, that have been largely summed up through written descriptions that in parts can be quite obscure, confusing and difficult to follow. I strongly suspect some of my concerns below would be addressed through greater clarity of explanation, model figures and maybe some extra technical detail in the supplementary information.

Main concerns

1. The calculation of the hospitalisation risk is very confusing in the manuscript. The is obviously a quantity that is highly heterogeneous over age and with the lack of age dependence the authors are correct to try and consider the constitution of the different regional populations. The authors use age-specific estimates for hospitalisation risk from literature and then, for each region, uses a weighted average of these. The text (in the first paragraph of the section) seems to suggest that the weighting is derived based on demography, before, in the second paragraph of the section, suggesting that the weighting is based on the age distribution of the cases. This latter weighting seems to make more sense, but I found these two paragraphs to be contradictory. Could it be different treatment of the symptomatic and asymptomatic infections?

2. Fitting of the regional model. The authors appear to make multiple use of the data, estimating the 'static' parameters (d, pi1, pi2, amplification) based on the national model before then disaggregating the data and then estimating the transmission parameters in the regional model assuming a prior for these parameters based on the first stage posterior. This appears, as written, to be a multiple use of the data overweighting the information it contains. Is this a fair reflection? Is there not some way that the data could be partitioned so that a double use of the data is not being made (i.e. estimate one based on hospitalisations, and the other based on test positive cases)?

3. The time from transmission to positive test is a prior a uniform rv over the range [0,4] days. Is this not VERY short, considering the expected incubation period a priori is 5.1 days? Would suggest that most people are testing positive before they've even got symptoms. Can this be so? Would it not be better to count d as the mean time between transmission and test positive and have some kind of convolution over this distribution (it doesn't have to introduce extra stochasticity) and would smooth out the projected curve of case counts.

4. The algorithm doesn't say what happens if the simulated data fall outside of the epsilon thresholds. I think it should say that you keep simulating until something does lie within the specified bounds, but there is a missing else statement. I guess also the definition of the weights doesn't need the case r = 0.

5. As mentioned in the intro above, mathematical descriptions of some of the model features, such as the hospitalisation risk mentioned above, in the supplementary information would be very welcome.

6. I may have missed any discussion on the availability of data. I can see there is a github repository for the code, which, as ever, is welcome.

(Very) Minor comment

5. I felt the paper was overselling itself in describing a novel way of accounting for vaccination.. many works before have had time-dependent betas mopping up all sorts of fluctuating transmission behaviour etc...

Reviewer #3: This is a rather long manuscript detailing (a) the use of a multi-region SEIR model for COVID-19, (b) how the authors overcome the parameter estimation problem by using a sequential approximate Bayesian computation approach, and (c) how they went on to use the methods on modeling roughly the first year of COVID-19 pandemic in Norway. The model described here seems interesting and useful, especially on the part where they incorporate mobility information. The manuscript went through great details on how to estimate parameters from the data for their models and make the model works, which honestly is a tough part of adding complexity to infectious disease model. I believe this manuscript provides significant methodological insight and will serve as a useful resource for future modelers. However, I can't recommend the manuscript for publication in its current form, and I believe a major revision of the manuscript is needed here, due to the poorly defined objectives, weak organization of the paper and communication about the model.

The manuscript is rather lacking in clear objectives. It is hard to know what is the novel part that you introduced to your model, based on the writing. Is the manuscript about making a multi-region SEIR model and that regions are interacting with each others and yet each region is separately parameterised? Is the manuscript about the "Split-SMC-ABC" that is particularly important in estimating parameters? Is the manuscript about showing how various regions in Norway experienced reduction of R values? Without clear objectives and questions that you are trying to answer, the results section looks like a mishmash of findings without clear thread and message. For example, why should the reader care about 3-week ahead prediction while also looking at how the R values have changed over time? Are the changes in R values over time in Norway going to help me understand if the model is a good choice? What can I learn from the regional mobility time series in relation to the objectives? I think the abstract is much better in framing the objectives than the manuscript itself. And I think you should organize the story around the objectives.

Although you talked about mobility data and how you analyse them at length, it wasn't clear to me how the mobility matrices were incorporated into the regional transmission model you formulated. You mentioned how people are moved from one region to another in the paragraph starting in Line 162, yet in the set of equations describing their local SEIR model, there was no mention about the movement of people among the locations. This is confusing, especially when I try to understand who are the people visiting other locality. Did you randomly move the people regardless of their state (S, E, I, R) and what are the justification for that? There is a lot of disconnect between the mobility part of the model with the local transmission part of the model.

As with the results section, the discussion also feels like a mishmash of ideas. You seem to interweave future direction, limitation, the good and bad about the model, and sometimes tangential findings or "tips" altogether. This is very well tied into the problem of lack of general sense of objectives. As a reader, I simply could not tell from the discussions if using the model is a good idea. In fact, I don't even quite understand what is exactly novel in the model already. I would again note that the abstract is better at selling the uniqueness of your model than the whole discussion section: "Our approach is the first regional changepoint stochastic metapopulation model capable of real time spatially refined surveillance and forecasting during emergencies." It doesn't help that the manuscript ends rather abruptly. This is a long manuscript and at times it helps to connect the part in the results to the part in the discussions, and this requires good organization of the paper.

Besides these major comments, I have some specific comments about the manuscript listed below:

I know that you're using mm/dd/yyyy, but I would recommend that dates be expressed in a less ambiguous way, e.g. Jan 1, 2020 or 2020 Jan 1.

Line 34: lowest *COVID-19* death rate

Line 102: There are 356 municipalities in Norway but 11 counties are modelled. It helps to explain what is municipality and what is county, also do you need to introduce the concept of municipality here since you're going to aggregate them into 11x11 mobility matrices anyway? Note also that in the manuscript in general, county is also known as region sometimes, perhaps it's wise to stick to one term?

Line 129: What's "churning" here?

Line 226: example for "changes in behaviour or viral properties"?

Line 232: Presumably, you mean "seeded the epidemic with *known* infected cases"?

Pg 12 and 13: I appreciate that you go into the details of how each of these data is extracted and analyzed or calculated, but would it help to be brief here and put the details in the SI? It helps to reduce the length and focus on your objectives.

Formula about Line 305: What is this quantity? What is t in this quantity?

Line 306: Assuming t means number of days since 2021 Jan 1. 9% increase occurs at around Jan 21 and 50% increase occurs at around Feb 12. These days are neither the center nor the end of the month, how did you choose these numbers to report?

Formula at Line 317: Nitpicky here but pi_0 and pi_1 would have meant detection probability at day 0 and day 1; I don't this is what you want?

Line 355 to 358: So you're comparing **final** observed hospital incidence and test to simulated ones? If so, it seems possible for the model to be completely wrong in the middle of trajectory but still get it right finally (e.g. wildly low incidence in the beginning and wildly high incidence in the end). Or did you compare monthly?

Line 456 to 457: Not sure what this sentence means, presumably you meant that 90% of hospitalisation occurs within 15 days of infection?

Line 523: 77% and 88% of pre-lockdown value?

Fig 3, 4: Explain the different shade of blues.

Line 534: Did you mean that you were using this model to publicly project case numbers, and people read your prediction and change their behaviour? If so, this is not conveyed the paper at all (that your predictions are perhaps being used by the government or media to communicate with the public.)

Table 1: "Bottom: Estimated reduction of transmissibility ... " as in reduced by percentage? or absolute reduction?

Line 548: What is appropriately coherent?

Table 1 and 2, Section in Line 570 onwards: What should I learn about this that is in line with your objectives?

Fig 5: All the other lines except for the green one is practically unreadable. Is there a need to code counties in numbers? Also, once again, is this presentation in line with the objectives?

Line 582 to 587: This section comes out of nowhere and is unclear which part is it tied to... is this related to estimates of pi_0 and pi_1?

Line 593 to 594: Why these five dates? And why are they not continuous?

Line 595 to 596: Might want to explain briefly what are energy score and CRPS? What number is considered good in general? Or is their sole purpose only to compare two models?

Line 642 to 644: Is this a "suggestion" or you did that in this work?

Line 645 to 656: Vaccination is almost not relevant in this work, and you did not mention it at all previously. Is this a "future direction"?

Line 657 to 660: How is it straightforward?

Line 729: Very helpful to mention this in your method (the SEIR model part for example) that one time step in your model is 6 hours. Confusing given that incidence and test data are in daily resolution, some of the formula seems to be in daily time steps too?

Line 746 & 749: Did you "show that" in the manuscript that prioritising visitors before host ... gives similar spatiotemporal dynamics? Did you "document that" your regularisation method retain the real characteristics? Or is this based on preliminary work? Not asking you to show/document them here just a poor choice of wording.

Reviewer #4: OVERVIEW

In this work, the authors report on incorporating high resolution mobility data into a mechanistic model of SARS-CoV-2 dynamics to inform near-real-time short-term projections of regional transmission and hospitalization data in Norway.

Overall, I think this work is an ambitious integration of data into a mechanistic framework, introducing a potentially interesting fitting approach, which is suited to the aims and audience of PLOS Comp Bio. However, I have several reservations about the work.

I'm roughly categorizing my concerns into what I'll summarily label mathematical (concerning the theoretical validity of the inference / calibration algorithms), engineering / operational (practicability / utility / suitability of the approach for purpose), and scientific (concerning data & potential for replicability and/or analysis using other theories). I realize those aren't the precise distinctions everyone would make here, just trying to give them a rough shorthand.

MATHEMATICAL CONCERNS

I find the Split ABC-SMC idea intriguing and it seems intuitively plausible that it could work as an inference technique. Here I'm defining "work" to mean asymptotically identifying best estimates, having intervals mean what their labels state, etc - the features that are demonstrated theoretically for various other ABC algorithms, but are not *necessarily* features of an arbitrary new tweak to using ABC. However, the manuscript provides neither theoretical or practical demonstration that this approach does those things. Personally, I'd prefer the practical version: using known model parameters, create a synthetic version of the kind of data you perform inference with, and characterize the performance of the method in recovering the known parameters.

I'd find that kind of demonstration sufficient, but in general the manuscript needs more evidence about how this method performs in abstract when the "answer" is known.

Related to this issue: a few places in the manuscript refer to picking between ABC targets, noting that hospitalization only resulted in higher variance. Seems like variance practically translates to uncertainty in forecasts, which isn't necessarily inappropriate. If the authors think that additional uncertainty is inappropriate, they could argue that from e.g. coverage. They have calculated those values, as indicated in the supplement - though the performance of the forecasting method should defiitely be main text results. A substantial part of what's being reported in this work is the performance of the approach, not just the raw outcome predicitons it makes.

ENGINEERING CONCERNS

My read of the authors argument is that they are advocating for an approach to making short-term projections of rates of transmission generally and hospitalization specifically that will be useful for policy decision-making (e.g. the public health outcomes of policies on the spectrum of relaxing => maintaining => enhancing intervention activities) and/or logistical planning for health care capacity. I am using "approach" here to encompass roughly 1) what data to collect (regionally dis-aggregated, mobility + epidemiological count data, but also the assorted other "standard" epidemiological measures [like dwell times in conditions, conditional probability of outcomes] & national statistics like age distribution), 2) how to synthesize that data (i.e. with region-level-resolution compartment model, with interacting regions), calibrate it (split SMC-ABC procedure + hospitalization/test positive count metrics), and what to predict / how to evaluate that prediction.

Implicit in that advocacy is that readers should use this method (or something like it) and / or the authors were justified in using this method to provide estimates for decision-making. To operationalize that argument, people need to compare this approach to alternatives for producing the same outcomes from the same data (or alternative data with some sort of valuation comparing collection requirements). What immediately comes to mind for the authors stated target (short-term, regional hospitalization counts) would be something like the various options for Rt estimation and forecasting / projection of cases / hospitalizations / etc (e.g. EpiNow2 R package, ). At the very least, some sort of null model with zeroth (static rate of outcomes) or first order elements (static rate of change in rate of outcomes) should be used. This requirement is one of the (many) standards proposed for forecasting / projection papers summarized in EPIFORGE (https://midasnetwork.us/epiforge-2020-guidelines/), which reflects at least some consensus in the community.

I don't see in here a clear statement of how much additional predictive power is wrung from the additional complexity baked into this mechanistic model *for the target predictions*. I agree that such mechanistic complexity may be necessary for other kinds of predictions (e.g. projections about total incidence over a longer planning horizon) or for critical-but-rare edge cases (e.g. what will happen initially with introduction of a new variant). But I don't see sufficient argument here that this model complexity is justifiably better than alternatives.

While the manuscript describes several other latent parameters or estimates that come out of the model, the discussion doesn't generally take a position on the value of thinking of these as predictions with direct public health consequences or of scientific interest (i.e. to be validated by new measurements).

Aside: I also think that more complex models invite lots of criticism of specific subelements. It's tough to address the validity of those criticisms without detailed sensitivity work. For example, some elements may seem very different from what "actually" happens, but the lower detail may not matter or may be counterbalancing assumptions elsewhere in the model (made for practical reasons). In the last section, there will certainly be a few of those, but as I said - it's unclear if they're important or not.

SCIENTIFIC CONCERNS

The distinguishing feature of science from math is the wide, transparent publication of input & result data; both share a requirement to describe a model to translate from inputs to outputs.

In that sense, the key data elements here seem to be the mobility measurements, the case series, and hospitalization series. Readers will understand that the raw data cannot be indiscriminately shared given privacy implications. However, as a practical matter, I doubt the raw data can be realistically requested for anything short of a full blown replication, and (unfortunately) about the only way that sort of activity would have any professional value would be some sort of systematic replication attempt of many modelling results. I suspect by the time such could be planned and funded, the pertinent raw data sources here would have decayed to an unusable state (n.b. I'd bet this is also true of any other publications considered for such an enterprise).

Given that practical reality, I think the authors need to do more to make this work meet data availability standards for something to constitute scientific activity. I am going to suggest a rough solution I think would satisfy the epistemic requirements, but of course there may be others and I'm open to other ways of addressing this issue.

My read of the model itself is that it compares to a time series of case and hospitalization data that is sufficiently aggegrated (no age, sex; at the regional level rather than high resolution) to confer anonymization. That aggregated data set could be made directly available as part one of the repositories, along with the aggregation scripts to go from the raw format, in plain text (csv) format. It won't be a particularly large file, even in plain text - order thousand rows, order 10 columns.

Related, I should point out that it's not obvious that hospitalization data is even available because of the 1K files display issue in github - there are serious organizational issues with that repository (e.g. files could be subdirectoried into xls/EVENT/date.xls and csv/EVENT/date.csv to make it much more manageable, though I understand its structure is now a dependency in various analyses) - and I don't think it's appropriate to expect a reviewer (or reader) to jump through hoops to check that. The recommended solution (clone the repo) seems likely to be inappropriate for bandwidth, data limited LMIC settings.

I think the same character of recommendations should be reasonable for the aggregated, regularized mobility matrices. The manuscript describes dropping low count movements, and has aggregated them to lower resolution in time and space than they are in the raw data. If the "example" mobility data in the availability statement is in fact already what I'm suggesting, then it should be characterized as such.

Again: I am open to alternative solutions by the authors. There are probably constraints on what they can do with the data that haven't been made explicit in the manuscript, some of which may preclude what I've suggested (aside: the authors / journal may wish these constraints be made explicit in the data availability statement to calibrate expectations when contacting the teleco). That understood, I have tried to paint a clear target of what I think is appropriate data to work from in terms of following the tenets of scientific activity.

Distinct from dealing with the issue of providing appropriate data for study replication, I think there are methodological gaps in the description of getting from raw data to the aggregated input. For example, while I definitely wish more countries would use a unique personal identifier for this kind of data and applaud Norwegian authorities for attempting to do so, I doubt that collection went perfectly - so how are data issues identified and resolved?

My personal solution to this would be to a high level plain language description in the main text, a more detailed description mixing plain language, math formulae, and algorithmic pseudo code in the SI, and the actual cleaning / transforming code in a repository. As written, the manuscript neglects some key details (namely, the general scheme for data hygiene), but also feels a bit too verbose in the main text - much of what's there currently would be much more accessible as specific bite-sized chunks in SI sections. The code for cleaning / formatting / etc is not obviously available.

ASSORTED MINOR ITEMS:

- How might "regional" be portably defined? This analysis seems to define it in terms of admin 0 (nation) vs admin 1 (immediate subnational) - but for other settings (e.g. US and US states, some of which exceed Norway's national population) that division is very different from the one in this analysis.

- I'm unclear on the exclusion of certain negative tests? i understand it for certain analyses - it's like you're trying to consolidate "negative" episodes, but ambiguous how to translate "duplicate" negative tests from an individual testing episode into insights about unobserved prevalence / incidence.

- re mobility assumptions - why these? what alternatives were explored? do those alternatives make a difference in mobility estimates? some indications in discussion, but not much by way of how much they matter. Re the bit about home establishment, why not based on modal location or modal location at night over a long period? How do inferred populations based on mobile data home establishment compare to other data sources for relative populations?

- the "four" mobility matrices (121 on pg 6) - first mention? what four?

- Re mobility regularization: is population conservation really appropriate? I understand the practical value of it, and would be okay with conserving total population, but presumably, Norway has some seasonal shift in typical locations of individuals? Seems inappropriate to enforce detail balance at each time step.

- Regarding preferential returns of vistors to their home location: how does that comport with the disaggregated mobility data indications?

- Does SEIR status change movement behavior? S and R shouldn't presumably. Probably E also fine. But I? Undetected Is maybe, but presumably Norway had isolation rules for detected Is.

- for seeding, wording suggests importation of known metadata (i.e. infections designated as travel acquired). Is that available from the reference epidemic data?

- using smoothed average instead of actual values - so what if the epi data are noisy? Doesn't the fitting algorithm / distance function / sequential quantiling address that?

- The lack of age structure muddies insight from other work on that - the aggregate estimates of severity fraction with time leaves a lot of wiggle room in the model, which might be inadvertantly counterbalancing some estimate that the model makes.

- tables 1-2:

* the concrete numbers suggest a certainty that seems unwarranted

* this would be better as a visualization, particularly for highlighting how long the estimation periods where relative to each other (and relative to estimating a single R value for the period)

- For mobility metric comparisions: is it appropriate to compare reductions / changes relative to only prior periods of analysis? or should comparison be reductions relative to historical trend for those periods?

- Discussion implies extending the model framework to include reinfections is straightforward. That seems overconfident. While the structural addition of waning to the equations is well-specified, there are several options to choose amongst, AND the "model framework" should be thought of as including changes to the data injestion & the fitting targets as well. While the authors might not change the fitting targets, that is still a choice that entails as strong as assumptions as any other.

- Re split-ABC-SMC: I'm not sure I understand the bit about increased probability of accepting parameters from previously accepted trajectories? Is that an algorithmic choice, or just a reflection of how the posterior => prior for next step works?

- There's no little mention of deaths in here; I understand that the model isn't *for* that, but would be a useful independent validation of the transmission aspects.

- The novelty assertions seem either wrong or qualified in someway that I don't understand. There's other spatio-temporally explicit work, including using mobility data, that is making short-term forecasts of transmission / outcomes. I don't see the need to assert novelty / priority here - this is a potentially interesting entry in a difficult-to-get-right genre.

- Regarding fitting resource indications (one week, 1400 cores): what is run time for a "single" sample? how many samples does that total calibration time correspond to?

**Have the authors made all data and (if applicable) computational code underlying the findings in their manuscript fully available?**

Reviewer #1: Yes

Reviewer #2: **No: **See point 6 in the review. I can see that the code has been made freely available, as have the mobility matrices, but I can't see any of the count (cases or hospitalisations) in there, nor any discussion of how available they are. One suspects they may be official statistics and available (at best) only be request, but hopefully the authors will deal with it as part of the response.

Reviewer #3: Yes

Reviewer #4: **No: **The authors have made most of the relevant bits available; how to obtain the data or synthetic example data is obvious, as is where to get the disease model, but it's non-obvious where to find their fitting code. That appears to be in what the data availability statement refers to as example mobility data.

It is not obvious where the overall analysis flow appears. Again, this appears to be in the repo referred to by the data availability.

In terms of unambiguous citations, the authors should produce a release tag for the repositories or at least reference a specific commit. I also recommend that they view the repositories (that they control) as effectively supplementary materials submissions, and thus do a bit more organization in the top level READMEs to walk readers through what's where, how to use the repo, etc.

As noted in more general notes to authors, I do think there should be a different set of data available.

PLOS authors have the option to publish the peer review history of their article (what does this mean?). If published, this will include your full peer review and any attached files.

Reviewer #1: No

Reviewer #2: No

Reviewer #3: No

Reviewer #4: **Yes: **Carl A. B. Pearson
---

## [Decision Letter · Decision Letter 1]

23 Nov 2022

Dear Dr de Blasio,

Thank you very much for submitting your manuscript "A real-time regional model for COVID-19: probabilistic situational awareness and forecasting" for consideration at PLOS Computational Biology. As with all papers reviewed by the journal, your manuscript was reviewed by members of the editorial board and by several independent reviewers. The reviewers appreciated the attention to an important topic. Based on the reviews, we are likely to accept this manuscript for publication, providing that you modify the manuscript according to the review recommendations.

Some of the reviewers still have a few suggestions for further clarifications. Reviewer 4 also asks you to compare the Rt estimates from your approach to those from an existing R package (e.g. EpiNow2) for a few example regions, which should be reasonably straightforward to do.

Regarding the data availability issue raised by Reviewer 4: I understand that there are instances in which data cannot be shared, but the science is still important. To address the reviewer's concern, I think it would be sufficient to more clearly state the steps that would need to be taken and permissions needed to access the data and/or provide simulated data that could be used to validate the findings of the study.

Sincerely,

Virginia E. Pitzer, Sc.D.

Section Editor

PLOS Computational Biology

Alex Perkins

Associate Editor

PLOS Computational Biology

Some of the reviewers still have a few suggestions for further clarifications. Reviewer 4 also asks you to compare the Rt estimates from your approach to those from an existing R package (e.g. EpiNow2) for a few example regions, which should be reasonably straightforward to do.

Regarding the data availability issue raised by Reviewer 4: I understand that there are instances in which data cannot be shared, but the science is still important. To address the reviewer's concern, I think it would be sufficient to more clearly state the steps that would need to be taken and permissions needed to access the data and/or provide simulated data that could be used to validate the findings of the study.

Reviewer's Responses to Questions

**Comments to the Authors:**

Reviewer #1: The authors have carefully responded to all of my comments and most of the limitations of the work have been now appropriately discussed. I think the manuscript is now suitable for publication.

Reviewer #2: Review uploaded as an attachment

Reviewer #3: I am satisfied with the edits and the response to my comments. Good job! I have only one major concern regarding the time step update in the model, and a nitpicky comment here.

> ANSWER: We have clarified further the connection between the local transmissibility model and the mobility by explicitly defining N_i^t through the mobility matrices X_ij^t in the Local stochastic

epidemiological model section.

> In addition, we have added a specification that we move individuals regardless of their disease state. We chose to do this as we do not know the infection status of the individuals in the mobility data. We agree that it is reasonable to assume that symptomatic individuals might move less, in particular if the symptoms are severe. We have still chosen to move them randomly independent of disease status as one would otherwise have to make quite strict ad-hoc assumptions where we do not have any data. The added specification to the Regional transmission model section is: ”We moved individuals independently of their disease status. Another option would have been to decrease the probability of travel for e.g. the infectious symptomatic, however we have chosen not to do this, as it would require ad hoc assumptions on the size of reduction”.

I think it is fair to assume that people move among counties regardless of disease status. However, I still find the section regarding the time steps update unclear. The movement of people shouldn't have just affected the N_t^i (which affects only the contact rate). Wouldn't S^i(t), E^i(t), I^i(t) be affected by the inward and outward flux of people too? Let's say there are 10 symptomatic infectious person in location i at time t. In the next time step, let's say none of them recovered, and no new exposed person become symptomatic infectious locally then I^i(t+1) would have been 10 according to equation (8). However, 3 of them move to other counties, and only 1 move into the county from elsewhere, then I^i(t+1) should have been 8 instead of 10 and this has some implication in the infection next time step. The movement of people isn't reflected in the set of equations (5) to (9). Perhaps what you did was to first update the number of people in each SEIR compartment to account for movement among counties at the beginning of each time steps (or maybe at the end of each time step), then you use equation (5) to (9) to update the movement among compartments? It would be nice to describe this in a few lines, if you don't want to over complicate the equations?

> ANSWER: We have specified the sentence by changing it to: “In addition, we included further changepoints to capture other potential gradual changes in behaviour or viral properties like the

gradual takeover by the more transmissible alpha variant in winter 2021 and potential tiring of compliance with restrictions over time”.

Perhaps 'compliance fatigue' is a more commonly seen phrasing vs 'tiring of compliance' here.

Reviewer #4: I think the authors' reply is responsive to my concerns. I'm not sure those concerns are fully resolved, however.

On validity:

Regarding demonstration of Split ABC-SMC, the main text should indicate that readers may refer to the SI for a demo calibration of the method. The method should be clearly caveated to the effect of that it hasn't undergone the same sort of validation that other ABC-SMC methods have, so shouldn't be assumed to have the guarantees of those methods.

I think the practical considerations used to constrain the demonstration validation were basically reasonable, though I would prefer an "unlike" demonstration in addition to the "alike" demonstration used.

However, I'm not sure I agree that the method was actually showing convergence to the input values - some of the intervals, like Oslo R2, are definitely there, but others, like Oslo R1, definitely *aren't*. The authors need to grapple with that outcome a bit more frankly.

On suitability:

I think the case that a regionally disaggregated approach is approach is preferrable is very clear, and I understand the comparison between national / regional forecasts. However, I do not think I was sufficiently clear about comparison to simpler methods in my original notes. The authors should just use one of the packaged non-parametric Rt estimation / incidence forecasting tools on some of the regional / national data and compete those estimates with the estimates from their more detailed models. I'm emphasizing "some" here, because I agree that data restrictions (e.g. too low hospitalization counts) may make those tools unusable for some regions. But it seems entirely reasonable to run one of the packaged Rt methods against the hospitalization data for e.g. Viken and Oslo and score that kind of non-parametric estimate against this approach (as well as evaluating computational resources etc to achieve the forecast). For the particular questions this work intends to address, the non-parametric Rt approach is more like a standard analysis, and so the authors should be characterizing their approach in terms of what it does better (e.g. probably dealing with the low-count regions) vs worse than the packaged Rt approaches.

On data availability:

I appreciate the sharing constraints and the easy way to satisfy them, but don't find them particularly tolerable as a reader. If the non-sensitive data are the only result that's publicly available, then a version of this analysis that is based on those data is what is appropriate *for the scientific literature* (reports for public health decision-making can be made to alternative standard). I'll also highlight again that asking readers to sift the raw public data into the format used in the analysis isn't appropriate, and furthermore it seems that resource may not have suitable longevity based on the README at https://github.com/folkehelseinstituttet/surveillance_data. The authors should figure out how to incorporate the relevant snapshot of that data as used in their analysis, in a more stable repository.

I also do not find the mobility data approach convincing. The example data is imprecisely documented (README indicates a Norway_adj data object, running a run_Norway.R script - that's presumably referring to data/dNorway.rds and test/run_Norway_test.R). The mob_data field in that object seems to be movement counts; the manuscript describes censoring counts < 20, but there are counts <20 in that data (e.g. mob_data[[1]] seems to be about 30% n < 20).

In the end, I'd refer to the PLOS data policy - "you make available all data used to draw the conclusions in the manuscript". If there are data that you'd like to use to drive some result, but can't make available, then you can't use them. If that means having to figure out a deindentified version of the data and then doing a lower resolution version of the analysis or whatever, well, then those the results that can be presented in the open scientific literature. I understand there are exceptions, but I'm not convinced that this analysis couldn't be done in a way more amenable to releasing data.

On assorted minor comments:

I found these responses mostly satisfactory.

I will clarify my question: "Regarding preferential returns of vistors to their home location: how does that comport with the disaggregated mobility data indications?" The model description says individuals are assigned a home location, i. They then visit j in some time period. In the next time period, any trips from j to i are preferentially assigned to people with home location i, currently visiting j (and then assorted other priority rules). My question concerned how well the individual data matched that assumption - from your reply, I gather you don't actually have access to the data in a way that would allow you to make that assessment. Is it possible to put that question to Telenor?

Regarding SEIR changing movement behavior: to be clear, the assumption made is an equally ad hoc quantity (that probability of movement remains = 1). I recommend considering a different angle on this argument.

For averaging, estimating a time-averaged reporting probability, rather then day-scaled ones, is fine, for the practical reasons the reply highlights. I still don't think it makes sense to feed the estimation procedure adulterated data (by applying a smoothing window). Probably doesn't make much of difference in terms of resulting estimates, but its not like its computationally cheaper because it doesn't change data resolution. So seems like no reason to do it?

**Have the authors made all data and (if applicable) computational code underlying the findings in their manuscript fully available?**

Reviewer #1: None

Reviewer #2: Yes

Reviewer #3: Yes

Reviewer #4: **No: **I think the authors have done a better job of making code available and accessible, but the data fundamentally still aren't there.

PLOS authors have the option to publish the peer review history of their article (what does this mean?). If published, this will include your full peer review and any attached files.

Reviewer #1: No

Reviewer #2: No

Reviewer #3: No

Reviewer #4: **Yes: **Carl A. B. Pearson

Figure Files:

Data Requirements:

Reproducibility:

References:

---

## [Editor Report · Decision Letter 2]

8 Jan 2023

Dear Dr de Blasio,

We are pleased to inform you that your manuscript 'A real-time regional model for COVID-19: probabilistic situational awareness and forecasting' has been provisionally accepted for publication in PLOS Computational Biology.

Best regards,

Alex Perkins

Academic Editor

PLOS Computational Biology

Virginia Pitzer

Section Editor

PLOS Computational Biology

---

## [Editor Report · Acceptance letter]

19 Jan 2023

PCOMPBIOL-D-22-00231R2 

A real-time regional model for COVID-19: probabilistic situational awareness and forecasting

Dear Dr de Blasio,

I am pleased to inform you that your manuscript has been formally accepted for publication in PLOS Computational Biology. Your manuscript is now with our production department and you will be notified of the publication date in due course.

With kind regards,

Zsofia Freund
